# An excess of niche differences maximizes ecosystem functioning

Oscar Godoy [1✉], Lorena Gómez-Aparicio [2], Luis Matías [3], Ignacio M. Pérez-Ramos [2] & Eric Allan [4]

Ecologists have long argued that higher functioning in diverse communities arises from the niche differences stabilizing species coexistence and from the fitness differences driving competitive dominance. However, rigorous tests are lacking. We couple field-parameterized models of competition between 10 annual plant species with a biodiversity-functioning experiment under two contrasting environmental conditions, to study how coexistence determinants link to biodiversity effects (selection and complementarity). We find that complementarity effects positively correlate with niche differences and selection effects differences correlate with fitness differences. However, niche differences also contribute to selection effects and fitness differences to complementarity effects. Despite this complexity, communities with an excess of niche differences (where niche differences exceeded those needed for coexistence) produce more biomass and have faster decomposition rates under drought, but do not take up nutrients more rapidly. We provide empirical evidence that the mechanisms determining coexistence correlate with those maximizing ecosystem functioning.

[1] Departamento de Biología, Instituto Universitario de Investigación Marina (INMAR), Universidad de Cádiz, E-11510 Puerto Real, Spain. [2] Instituto de Recursos Naturales y Agrobiología de Sevilla (IRNAS-CSIC), LINC-Global, Av. Reina Mercedes 10, E-41012 Sevilla, Spain. [3] Departamento de Biología Vegetal y Ecología, Universidad de Sevilla, Av. Reina Mercedes 10, E-41080 Sevilla, Spain. [4] Institute of Plant Sciences, University of Bern, Altenbergrain 21, 3013 Bern, Switzerland. ✉email: oscar.godoy@uca.es

A large number of experimental and observational studies have shown that more diverse communities have higher levels of multiple ecosystem functions[1–5]. At the same time, global change is reducing opportunities for coexistence[6,7], and therefore diversity[8], ultimately reducing a range of ecosystem functions and services[9]. A better understanding of the connections between the processes maintaining biodiversity in communities and those driving functioning would allow us to better predict effects of global change on ecosystem functioning and to optimize restoration efforts. However, links between the conditions necessary for species to coexist and the processes driving high functioning in diverse communities have remained elusive.

Two main effects have been identified as underlying causes of biodiversity–ecosystem function relationships[10]: complementarity effects occur when species, on average, yield more in mixture than in monoculture, and selection effects occur when there is a covariance (positive or negative) between monoculture yield and dominance in mixture. Ecologists have often invoked principles from coexistence theory[11] to explain these effects. There is a general assumption that complementarity effects are driven by niche differences, which stabilize species coexistence when intraspecific competition exceeds interspecific competition[10,12,13]. It is also tempting to assume that selection effects are driven by species competitive abilities or fitness[14]. Surprisingly, empirical support for these connections is currently lacking because studies have not quantified selection and complementarity effects together with the determinants of competitive outcomes (i.e., niche and fitness differences). Both the two biodiversity effects and the two determinants of coexistence can be driven by multiple underlying mechanisms: complementarity and niche differences for instance can be driven by interspecific differences in resource use, the action of specialist natural enemies or facilitation, amongst other processes[15]. While it would be ideal to link all these underlying mechanisms together, i.e., to understand how differences in resource use, natural enemies, facilitation, etc. combine and interact to drive both community assembly and functioning, this will be highly challenging. However, showing how overall biodiversity-functioning and coexistence processes relate to each other under is a first step towards providing a unified view of the conditions needed for stable coexistence and the conditions for high ecosystem functioning.

Recent theory shows that the relationship between selection and complementarity effects and niche and fitness differences can be more complicated than initially assumed[11,16]. The main reason is that selection and complementarity effects are determined by species' relative abundances and by density-dependent effects, which emerge from the combination of niche and fitness differences between species[17–19]. For instance, complementarity effects arise when interspecific competition is reduced relative to intraspecific competition, which is also the definition of niche differences[11,16]. However, differences in competitive ability between species also influence their relative abundances and reduce evenness, which would be expected to reduce complementarity between them as well[11,20]. At the same time, negative selection effects arise when low-yielding species increase their functioning in mixtures, which could be driven by them experiencing lower inter than intraspecific competition and this should also drive niche differences between them. Given this interdependence of coexistence mechanisms[21], a question worth asking is what combination of niche and fitness differences maximizes functioning. In general, we expect that larger niche differences should promote complementarity effects and smaller fitness differences should reduce the relative abundance of the superior competitor and enhance evenness. Both of these processes should enhance functioning and our main hypothesis is therefore that the communities with the highest functioning will

be those in which both stabilizing and equalizing processes are operating. Stabilizing processes are those which enhance niche differences and have been the historical focus of coexistence research. The second, equalizing effects, are much less studied but arise when species equalize their response to both intra and interspecific competition (i.e., they become more similar in their overall sensitivity to competition). We expect that both processes will be operating in high-functioning communities, and the highest functioning should arise where niche differences are stronger than those required to maintain coexistence, i.e., in communities with an excess of niche differences. We can contrast this with the situation in which only stabilizing or equalizing processes matter and therefore in which niche differences or fitness differences alone, not the excess of niche differences, maximizes or minimizes function. Comparing these two possibilities will allow us to determine the link between the determinants of competitive outcomes and ecosystem functioning.

Coexistence and biodiversity processes are also likely to vary depending on environmental context. Several studies have shown that biodiversity-functioning relationships are modified by a range of environmental factors such as stress or resource levels e.g., ref. [22]. Niche and fitness differences between species are also modulated by environmental conditions, such as water availability[7,23]. Because of this context dependency, determining when links between stable coexistence conditions and high functioning vary with environmental conditions may help to determine if there are circumstances under which positive complementarity effects can occur even without stabilizing niche differences[24,25].

To rigorously evaluate the relationships between biodiversity-functioning and coexistence processes, we performed a combined competition and biodiversity-functioning experiment with ten annual plant species (Table 1) in a Mediterranean grassland, which allowed us to field-parameterize population models to quantify stabilizing niche differences and average fitness differences. In this experiment, we directly manipulated the timing of species germination to create two contrasting scenarios of water availability (control climate and drought treatments; see the section "Methods" for more details). Our manipulation modifies the niche and fitness differences between species pairs[7] and we expect that complementarity and selection effects would also be modified by this environmental variation. The drought treatment therefore allows us to estimate how changes in coexistence mechanisms modulate the net effect of diversity on productivity. Finally, it is worth highlighting that selection and complementarity effects have almost always been assessed for biomass, but other critical ecosystem functions may show different responses to biodiversity[1]. We therefore also aim to test whether the conditions promoting high stable coexistence also promote high levels of functions other than biomass such as litter decomposition or soil nutrient cycling.

Here, consistent with recent theoretical advances[11,18], we provide empirical evidence that both selection and complementarity are related to a combination of the stabilizing niche differences that promote species diversity and to the average fitness differences that promote competitive exclusion. Despite these complex relationships, we find following our predictions that more stable coexistence promotes higher biomass. This implies that any process that destabilizes coexistence should reduce biomass and in fact it may be reduced even before coexistence is threatened. However, we also detect that extending these findings to functions beyond biomass is not straightforward because they are likely driven by other mechanisms unrelated or poorly related to plant coexistence. These discrepancies call to further develop a framework to link the species differences allowing stable coexistence of several trophic levels to those

**Table 1 Species and assembled communities with the diversity levels used in the experiment.**

| Species | Family | Code | Community | Composition |
|---|---|---|---|---|
| *Bromus madritensis* | Poaceae | BRMA | 3 species | 1 (BRMA, BOOF, MACA), 2 (BOOF, CABU, MEPO), |
| *Borago officinalis* | Boraginaceae | BOOF | | 3 (CAAR, PARO, VISA), 4 (BRMA, CABU, MEPO), |
| | | | | 5 (CABU, DIER, MACA), 6 (BRMA, DIER, SIAL). |
| *Calendula arvensis* | Asteraceae | CAAR | 5 species | 1 (BRMA, CAAR, CABU, DIER, MEPO), |
| *Capsella bursa-pastoris* | Brassicaceae | CABU | | 2 (BOOF, CAAR, DIER, PARO, VISA), |
| | | | | 3 (BOOF, CABU, DIER, PARO, VISA), |
| | | | | 4 (BRMA, BOOF, MACA, MEPO, VISA). |
| *Diplotaxis erucoides* | Brassicaceae | DIER | 7 species | 1 (BRMA, CABU, DIER, MEPO, SIAL, PARO, VISA), |
| *Matricaria chamomilla* | Asteraceae | MACA | | 2 (BRMA, CAAR, MACA, MEPO, PARO, SIAL, VISA), |
| | | | | 3 (BRMA, BOOF, CAAR, CABU, DIER, MACA, PARO). |
| *Medicago polymorpha* | Fabaceae | MEPO | 9 species | 1 (BRMA, BOOF, CAAR, DIER, MACA, MEPO, PARO, SIAL, VISA), |
| *Papaver rhoeas* | Papaveraceae | PARO | | 2 (BRMA, CAAR, CABU, DIER, MACA, MEPO, PARO, SIAL, VISA). |
| *Sinapis alba* | Brassicaceae | SIAL | 10 species | 1 (BRMA, BOOF, CAAR, CABU, DIER, MACA, MEPO, PARO, SIAL, VISA). |
| *Vicia sativa* | Fabaceae | VISA | | |

promoting high levels of different functions. Taken together, our results provide a first step in this direction by showing that the conditions promoting stable coexistence and high ecosystem functioning within a trophic level are the same.

## Results

**Diversity-functioning relationships**. We first analyzed the overall relationship between diversity and function and found that more diverse communities produced more biomass, their litter decomposed faster and they took up more soil N than less diverse communities, although the magnitude of these relationships depended on the climatic conditions (Fig. 1). These positive diversity effects mostly resulted from an increase in complementarity effects with increasing community diversity. In contrast, selection effects became more negative with increasing diversity (Supplementary Fig. 1).

**Mapping of coexistence and biodiversity mechanisms**. We next related biodiversity-functioning to coexistence mechanisms. Niche and fitness differences are defined for pairs of species[18] in the annual plant model (see Eqs. 2 and 3, "Methods"), but complementarity and selection effects are commonly measured at the community level[10]. To compare the effects, we therefore adapted diversity interaction models[20] to calculate measures analogous to complementarity between pairs of species and selection effects for individual species, which we converted to pairwise differences in selection effects. When averaged at the community level, these measures of selection and complementarity effects were generally similar to those obtained from the additive partition of Loreau and Hector[10] (Supplementary Fig. 2). For all functions evaluated, pairwise complementarity effects were higher when stabilizing niche differences were large (significant for biomass and for soil N under drought) and when average fitness differences were small (significant for litter decomposition under drought). In contrast, pairwise differences in selection effects were larger when niche differences were small (significant for soil N under drought) and generally when fitness differences were large (significant for biomass and for soil N under drought), although the opposite trend was found for soil N under control conditions (Fig. 2). Although the direction of effects was generally consistent across functions, their significance varied, and complementarity effects were sometimes only weakly linked to niche differences (Fig. 2). Surprisingly, these relationships across functions were in general stronger under drought conditions, despite the fact that the drought treatment

significantly reduced both niche and fitness differences[7] (Supplementary Fig. 3).

Fitness differences between species can result from differences in demography (i.e., differences in intrinsic growth rates), or from differences in competitive response (i.e., differences in the sensitivity of species to competition; see "Methods", Eq. 3). This means that a species can be a good competitor because it produces many seeds, because its seed production is barely reduced in the presence of competitors, or by a combination of both processes. In order to investigate the importance of these two components, we correlated each one with complementarity and selection and found that they did not contribute equally to observed complementarity and selection effects. Complementarity effects only correlated significantly with the competitive response ratio, not the demographic ratio, and only under the control climate (Fig. 2). This suggests that asymmetries in species' sensitivity to competition, rather than differences in their growth rates, reduced complementarity effects between them. In contrast, both demographic and competitive response differences were correlated with differences in selection effects. Demographic differences correlated with differences in selection effects on biomass (control climate) and soil nitrogen content (drought) and competitive response differences correlated with differences in selection effects for soil nitrogen content and litter decomposition under drought (Fig. 2). Importantly, the relationships observed for soil N were the same for the other soil elements analyzed, namely total organic carbon, available P, and exchange cations (Supplementary Fig. 4).

**What combination of niche and fitness differences maximizes biomass?** Both stabilizing niche differences and average fitness differences influenced multiple ecosystem functions. We therefore evaluated their combined effect on functioning, i.e., to test whether species pairs that are predicted to more stably coexist have higher functioning. Supporting our main hypothesis, we found that the species pairs predicted to coexist most stably (i.e., those in which observed niche differences most strongly exceeded the minimum required to offset fitness differences, hereafter the *excess of niche differences*), were in turn predicted to produce significantly more biomass under both climatic conditions (Control climate Mantel $r = 0.40$, $P = 0.026$; Drought Mantel $r = 0.46$, $P = 0.012$), and to have faster litter decomposition under drought (Mantel $r = 0.29$, $P = 0.047$). However, they were not predicted to have higher levels of the other functions, i.e., litter decomposition under control climate (Mantel $r = 0.10$, $P = 0.462$) or soil nutrient content (Control climate Mantel $r = -0.15$,

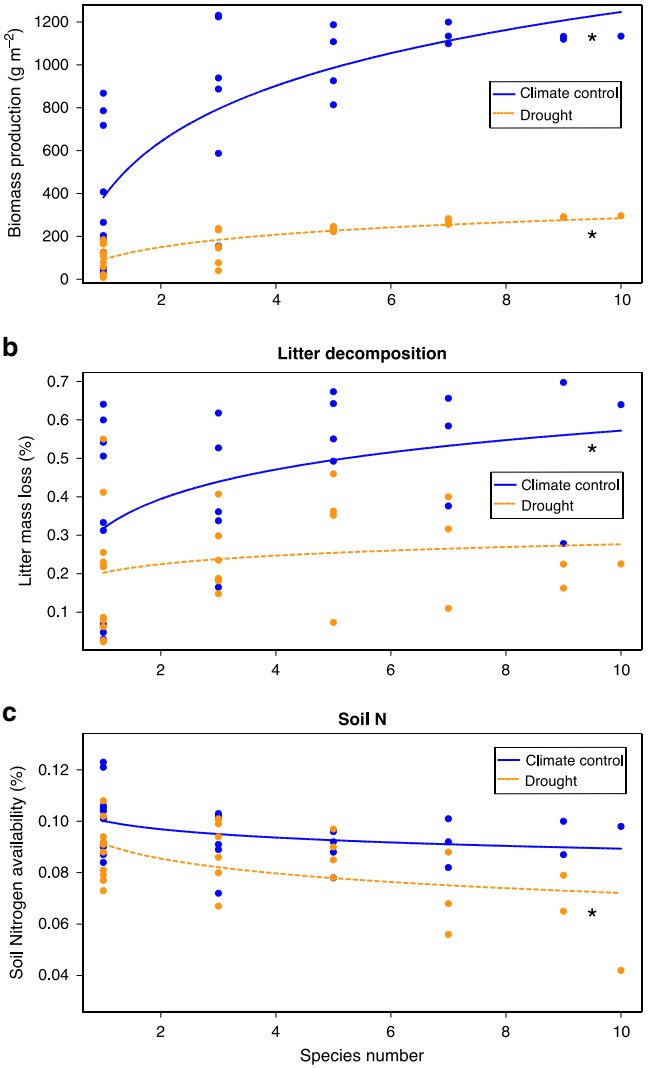

**Fig. 1 Observed effects of species diversity on biomass production, litter decomposition, and soil nutrient availability.** Panels from **a**–**c** include information of biomass production, litter decomposition, and soil N availability respectively. Total Soil N is shown here as an example of soil nutrient content but very similar relationships were observed for the other soil elements measured (C, P, Ca, Mg, and K). Blue lines and points represent the communities under control climate conditions and orange dashed lines and points show the same communities under drought conditions. Nonlinear instead regressions fitted the data better. Significant regressions at $P < 0.05$ are represented with an asterisk.

$P = 0.786$; Drought Mantel $r = -0.18$, $P = 0.831$) (Fig. 3). To test whether niche differences alone, fitness differences alone, or the combination of both measured as the excess of niche differences promoted higher functioning, we correlated both types of species differences with functioning. We found that the excess of niche differences better predicted biomass of species pairs than niche differences or fitness differences alone, respectively (95% confidence interval did not include zero; Mantel correlations for niche differences alone with biomass Control climate $r = 0.15$, $P = 0.227$; Drought $r = 0.21$, $P = 0.183$, Mantel correlations for fitness differences alone with biomass Control climate $r = -0.23$, $P = 0.141$; Drought $r = -0.28$, $P = 0.063$). For litter mass loss under drought, the excess of niche differences only marginally significantly correlated with function (90% confidence interval

did not include zero; most likely due to a lack of statistical power; Fig. 3). This last result suggests that combining niche and fitness differences better explains biomass than niche differences or fitness differences alone.

## Discussion

Understanding connections between the factors that promote species coexistence and high ecosystem functioning would allow a better mechanistic understanding of how biodiversity loss translates into reductions in different ecosystem functions. Both fields have developed frameworks to unify multiple underlying mechanisms into overall classes, but theoretical attempts to link the frameworks have shown that they cannot be easily mapped onto each other. However, by combining recent advances in coexistence theory with a series of competition and biodiversity-functioning experiments, we could show that the two frameworks can be linked and that positive effects of biodiversity on functioning resulted from a combination of large niche differences stabilizing coexistence, and small fitness differences equalizing competition between species. Moreover, our results provide a clear link between the conditions for stable coexistence and high functioning by showing that biomass is maximized when species coexist more stably, i.e., when niche differences more strongly exceed fitness differences (an excess of niche differences).

According to the classical expectations, we provide empirical support for the assumption that niche differences underlie complementarity, while large competitive ability differences result in large differences in selection effects. However, our results also support recent theoretical suggestions[11,16] that both selection and complementarity effects include a combination of niche and fitness differences (Fig. 2). We generally found larger complementarity effects when species differed in their niches. The niche differences and complementarity effects could be driven by many underlying processes such as species differences in resource uptake or responses to natural enemies. Facilitation between species can also promote complementarity[26] but seems to be of minor importance in our experiment (see "Methods"). Larger differences in selection effects were found when species differed substantially in fitness. Depending on the ecosystem function and environment, differences in selection effects could be driven, either by differences in species intrinsic growth rates, or differences in their response to competition. These results imply that functioning should be driven by a smaller number of species (highly positive selection effects) when communities contain species that vary more in growth rate (e.g., differences among species in resource conservation versus acquisition)[27], or under conditions that enhance differences in competitive ability, such as high nutrient availability[28].

Complementarity effects were promoted by niche differences but were also reduced when species differed strongly in fitness, more specifically, if they differed in their response to competition. In addition, differences in selection effects between species were reduced when they differed strongly in their niches. The negative effects of fitness differences on complementarity, and the negative effects of niche differences on differences in selection effects, are likely due to density-dependent processes, which simultaneously affect both niche and fitness differences (see definitions of Eqs. 2 and 3). The lack of independence between stabilizing and equalizing mechanisms has been recently acknowledged[21], and it illustrates how two interrelated processes should occur at the same time to increase function. First, fitting with the classic expectation, species should differ in their niches, but second and much less intuitive, species should have similar sensitivities to competition. Differences in how plant species acquire basic resources for growth, such as light, water, and carbon, may drive

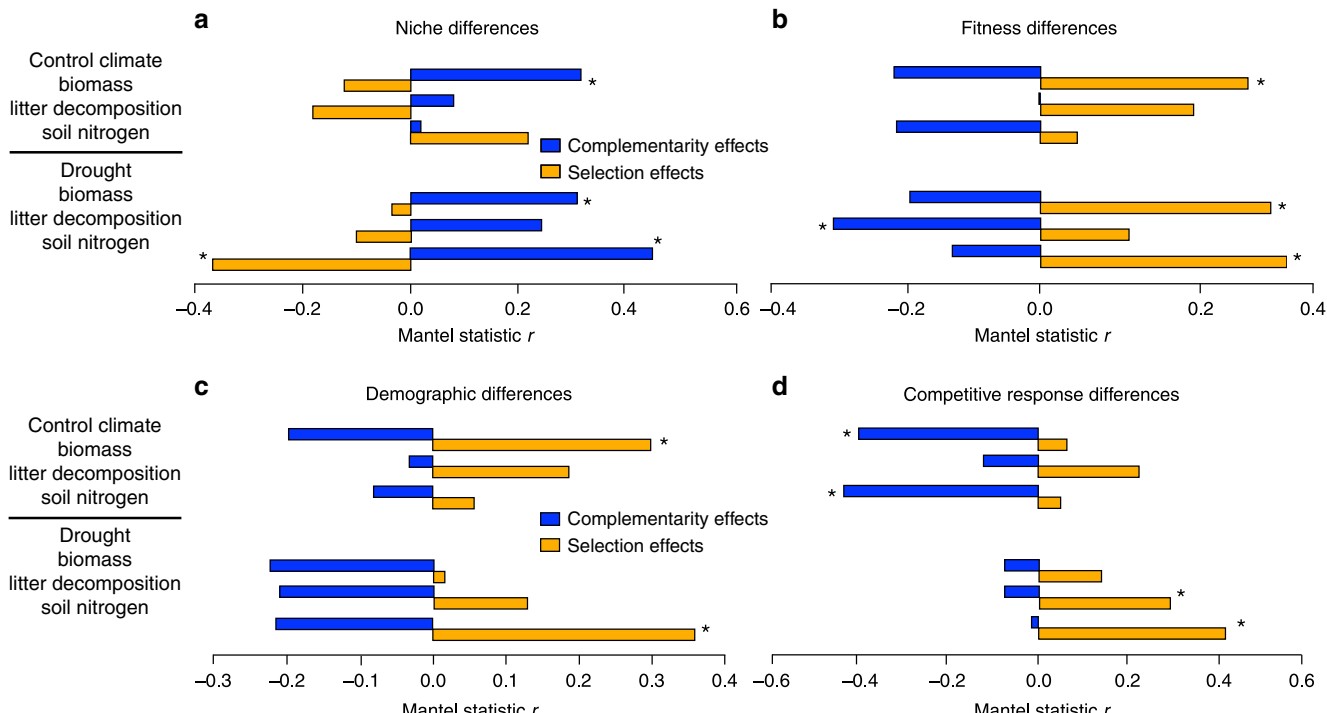

**Fig. 2 Correlations between stabilizing niche and average fitness differences and complementarity and selection effects.** Correlations are shown for the three functions considered, under the two contrasting environmental conditions (control climate, drought). Correlations between complementarity (blue) and selection (orange) and niche are fitness differences are shown in panels **a** and **b**. They are also shown with the two components of fitness differences, the demographic ratio (**c**) and the competitive response ratio (**d**). Significant correlations following a two-sided test are marked with an asterisk. To assess the degree of statistical significance while controlling for false discovery rate, we performed Benjamini–Hochberg corrections on raw P values for multiple comparisons ($n = 24$ comparisons per species difference evaluated).

both the niche differences and species sensitivity to competition. For instance, we observed that light curve convexity (a physiological trait representing the nonlinear saturation of photosynthetic activity with light availability) contributes to both niche differences and competitive response differences under drought conditions, in our system. On the other hand, water use efficiency (measured by carbon isotope ratio) seems to contribute to niche differences while differences in seed mass contribute to competitive response differences[29]. More research is needed to uncover all the underlying mechanisms that are important in driving niche and competitive response differences. However, our results show that, contrary to previous expectations, it is not just processes that drive niche differences between species but also those that equalize response to competition that are important in driving complementarity effects.

Given this more complex relationship between the determinants of coexistence and the mechanisms promoting positive effects of biodiversity on functioning, we also conducted an integrated analysis to determine if those communities where coexistence was most stable had highest functioning. In support of our main hypothesis, we found that more stable pairs (i.e., those in which niche differences more strongly exceeded fitness differences) were predicted to produce significantly more biomass (and there was a trend for litter to decompose faster under drought conditions) (Fig. 3). These results resemble prior theoretical findings that biomass is directly associated with the niche differences between species[11,16], yet our results provide insights into a subtle but important difference. Here we show that it is the difference between observed niche differences and the minimum niche differences necessary for coexistence, rather than niche differences alone, that is critical for high functioning,

i.e., an excess of niche differences leads to the highest biomass production. This result therefore provides a clear link between the conditions required for stable coexistence and those promoting high ecosystem functioning and shows that a combination of stabilizing and equalizing processes leads to highest productivity. Such link is illustrated in our system by the species pair *Vicia sativa* (Fabaceae) and *Borago officinalis* (Boraginaceae) (Table 1). Both species maximize stabilizing and equalizing process by two contrasting functional strategies. Individuals of *B. officinalis* have, on average, lower seed mass and leaf dry matter content but higher water use efficiency than *V. sativa*[29]. These differences results in a large niche difference and low fitness difference between the two and likely contributes to the high predicted biomass (120 g m$^{-2}$) of the species pair under climate control. Similarly, *Calendula arvensis* (Asteraceae) and *Bromus madritensis* (Poaceae) differ strongly in light curve convexity under drought conditions, which results in a large niche difference and high biomass for the species pair. The precise mechanisms enhancing coexistence are likely to differ between species pairs but an excess of niche differences seems to consistently promote functioning, meaning that a reduction in stabilizing or equalizing processes, even if the reduction is not strong even to prevent coexistence, may still reduce ecosystem functioning.

Although they were productive, more stably coexisting plant communities did not show faster litter decomposition rates under control climate conditions, or higher soil nutrient uptake across our experimental treatments (Fig. 3). These results suggest that the conditions leading to more stable coexistence of plant species mostly maximize functions directly related to plant performance, such as biomass, but their effects are less pronounced on

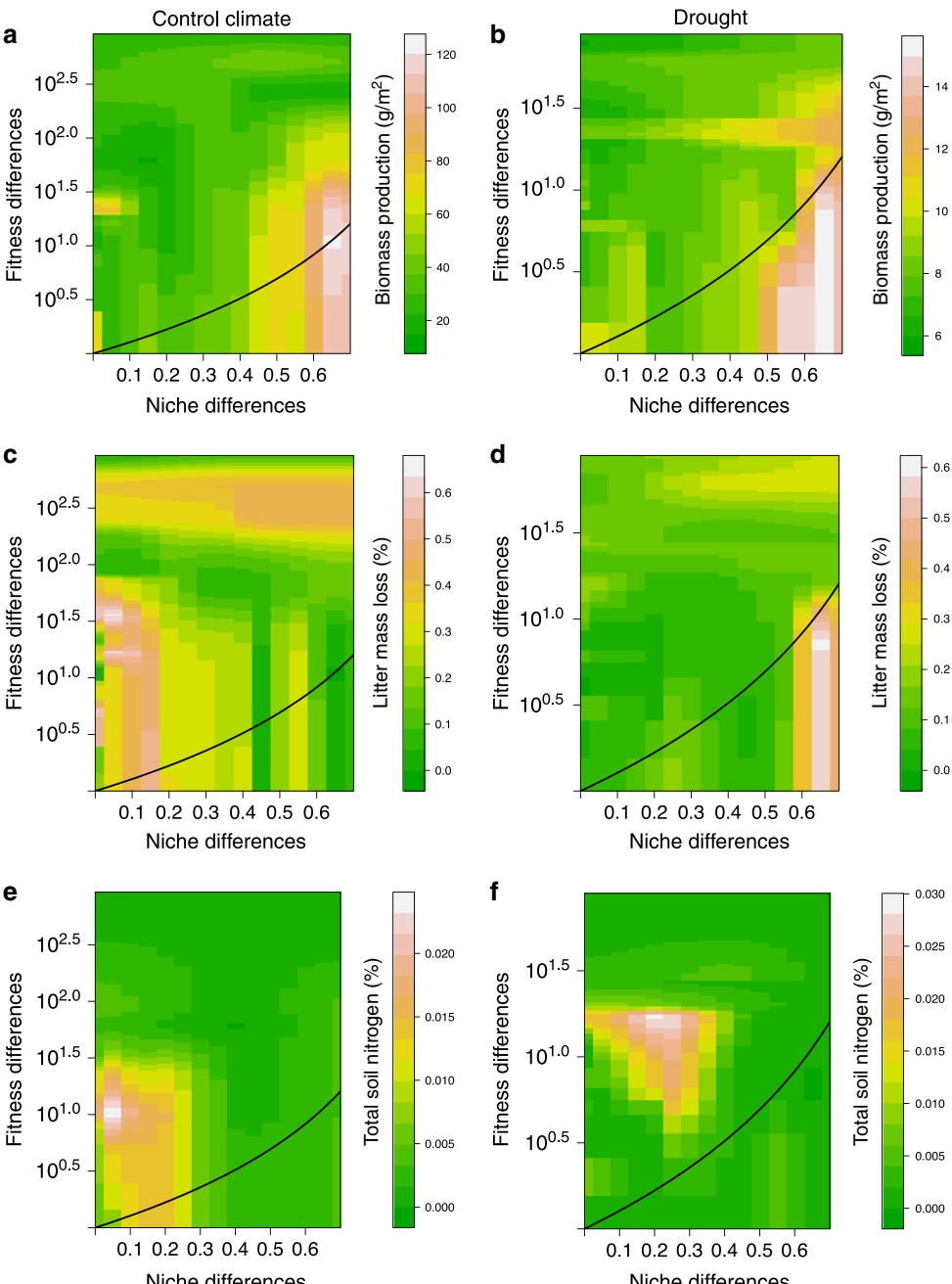

**Fig. 3 Relationship between the degree of stability of coexistence and the levels of multiple functions.** Biomass, upper two panels (**a** and **b**); litter decomposition, middle two panels (**c** and **d**); total soil N, lower two panels (**e** and **f**) under control climate conditions (left panels), and drought (right panels). The heat map represents the level of functioning estimated for a given species pairs, using the approach of (20). Greener colors represent low functioning while brownish to white colors represent higher functioning. The solid black line indicates whether the conditions for coexistence are met ($\rho < \frac{\kappa_j}{\kappa_i}$, where species $j$ is the fitness superior) and separates the coexistence from the competitive exclusion region. Mantel tests, following Benjamini–Hochberg correction for multiple comparisons, showed that the excess of niche differences for a species pair (i.e., their distance from the coexistence line) was significantly related to their predicted biomass and the predicted litter decomposition under drought but not under control climate conditions or soil nitrogen availability. For a graphical representation of observed pairwise niche and fitness differences (Supplementary Fig. 3).

functions that are indirectly related to performance. A potential explanation of these mismatches is that these other functions more strongly involve the effect of other trophic levels. For instance, litter decomposition is influenced not only by leaf litter traits, but also by the effect of soil organisms, including macro- and micro-invertebrates, nematodes, bacteria, and fungi[30]. In addition, the combination of plant traits that leads to high litter decomposition, or soil nutrient uptake, may be different from those determining stable coexistence and high performance.

These results show that linking coexistence and functioning is likely to be more complex for functions other than biomass. To understand how diversity loss affects these functions we may need first to consider the mechanisms promoting coexistence not only of plants but of multiple trophic levels[31], and once established these mechanisms, we need to test to what extent they relate to multiple functions. Performing these two steps is necessary because there might be a subset of functions that are not linked or poorly related to coexistence mechanisms.

Despite quantifying both coexistence mechanisms and biodiversity effects for multiple functions requires a considerable effort, we could not include the spatial and temporal variation that is key to maintain diversity at larger scales[32]. In our approach only coexistence mechanisms that operate in constant environments can contribute to the niche differences we measured[14]. Nevertheless, we do find a significant link between stable coexistence and biomass, which suggests that nonspatial/temporal coexistence mechanisms such as resource partitioning or natural enemies do promote high biomass in this system. Evaluating how much ecosystem functioning is additionally provided by coexistence mechanisms operating in variable environments is a promising direction for further research.

Environmental conditions affected the strength of relationships between coexistence and biodiversity mechanisms. In our study, delaying germination decreased rainfall by almost 40% and reduced the growing season by 2 months. This delay strongly reduced biomass to about 10% of the level in control conditions (Fig. 1), consistent with the predominant role of water availability in controlling biomass yields in Mediterranean ecosystems (e.g., ref. [33]). This reduction in water availability might be expected to reduce available niches and competitive ability differences and we did find evidence that wetter environmental conditions allowed for greater niche and fitness differences between species pairs (Supplementary Fig. 3)[7]. With greater niche overlap, it is reasonable to expect a weaker relationship between stabilizing niche differences and complementarity, however, we actually observed a stronger correlation between them (Fig. 2). This implies that not all of the processes driving niche differences contributed to complementarity effects under wet conditions (Fig. 2). Our approach is phenomenological, which means that we do not know the specific sources of variation in observed niche differences in our experiment. Nevertheless, these results emphasize the context-dependency of biodiversity effects on functioning and call for a framework to understand what type of environmental conditions promote the niche differences, and differences in species sensitivity to competition, that contribute to complementarity effects.

Our study represents a step forward in evaluating the link between the drivers maintaining diversity and functioning compared to previous experimental work that considered particular components (e.g., interspecific facilitation[34]) or aggregates (e.g., community evenness[35]) of niche and fitness differences. Still our approach to measure community stability is fundamentally based on pairwise interactions between species. The next step is to move beyond this pairwise framework to one in which niche and fitness differences are estimated at the community level, and in which additional factors determining coexistence at the multispecies level such as indirect or higher-order interactions are incorporated[36]. Although recent toolboxes have been proposed[37], we lack clear expectations about how the mechanisms determining the degree of stability in complex communities are linked to the net effect of biodiversity on functioning. However, incorporating these effects would be important to test if the communities providing high levels of functions like those related to nitrogen cycling are in fact able to stably coexist. Multispecies interactions under a multitrophic perspective may be more important for explaining other functions or functioning in other contexts but our results suggest that, for biomass at least, the conditions promoting stable coexistence for species pairs and high ecosystem functioning are the same.

## Methods

**Study site and experimental setup.** Our experiment was conducted at the La Hampa field station of the Spanish National Research Council (CSIC) in Seville, Spain (37°16′58.8″ N, 6°03′58.4″ W), 72 m above sea level. The climate is Mediterranean, with mild, wet winters and hot, dry summers. Soils are loamy with pH = 7.74, C/N = 8.70 and organic matter = 1.16% (0–10-cm depth). Precipitation totaled 532 mm during the experiment (September 2014–August 2015), similar to the 50-y average. We used ten common annual plants, which naturally co-occur at the study site, for the experiment. These species cover a wide phylogenetic and functional range and include members of six of the most abundant families in the Mediterranean grasslands of southern Spain (Table 1). Seeds were provided by a local supplier (Semillas silvestres S.L.) from populations located near to our study site. Our experiments were located within an 800 m² area, which had been previously cleared of all vegetation and which was fenced to prevent mammal herbivory. Landscape fabric was placed between plots to prevent growth of weeds.

**Theoretical background for quantifying niche and fitness differences.** Here we summarize the approach developed in ref. [38] to quantify the stabilizing effect of niche differences and average fitness differences between any pair of species. Both these measures are derived from mathematical models that capture the dynamics of competing annual plant populations with a seed bank[19,39]. This approach has been used in the past to accurately predict competitive outcomes between annual plant species[38]. Population growth is described as:

$$\frac{N_{i,t+1}}{N_{i,t}} = (1 - g_i)s_i + \frac{\lambda_i g_i}{1 + \alpha_{ii}g_i N_{i,t} + \Sigma_{j=1}^{S}\alpha_{ij}g_j N_{j,t}}, \qquad (1)$$

Where $\frac{N_{i,t+1}}{N_{i,t}}$ is the per capita population growth rate, and $N_{i,t}$ is the number of individuals (seeds) of species $i$ before germination in the fall of year $t$. Changes in per capita growth rates depend on the sum of two terms. The first describes the proportion of seeds that do not germinate $(1 - g_i)$ but survive in the seed soil bank $(s_i)$. The second term describes how much the per germinant fecundity, in the absence of competition $(\lambda_i)$, is reduced by the germinated density of conspecific $(g_i N_{i,t})$ and various heterospecific $\left(\Sigma_{j=1}^{S}g_j N_{j,t}\right)$ neighbors. These neighbor densities are modified by the interaction coefficients describing the per capita effect of species $j$ on species $i$ $(\alpha_{ij})$ and species $i$ on itself $(\alpha_{ii})$.

Following earlier studies[14,38], we define niche differences $(1 - \rho)$ for this model of population dynamics between competing species as:

$$1 - \rho = 1 - \sqrt{\frac{\alpha_{ij}}{\alpha_{jj}}\frac{\alpha_{ji}}{\alpha_{ii}}}. \qquad (2)$$

The stabilizing niche differences reflect the degree to which intraspecific competition exceeds interspecific competition. $1 - \rho$ is 1 when individuals only compete with conspecifics (i.e., there is no interspecific competition) and it is 0 when individuals compete equally with conspecifics and heterospecifics (i.e., intra and interspecific competition are equal). Niche differences between plant species can arise for instance from differences in light harvesting strategies[29,37–39], or in soil resource use and shared mutualisms[40].

The average fitness differences between a pair of competitors is $\frac{\kappa_j}{\kappa_i}$[38], and its expression is the following:

$$\frac{\kappa_j}{\kappa_i} = \frac{\eta_j - 1}{\eta_i - 1}\sqrt{\frac{\alpha_{ij}}{\alpha_{ji}}\frac{\alpha_{ii}}{\alpha_{jj}}}. \qquad (3)$$

The species with the higher value of $\frac{\kappa_j}{\kappa_i}$ (either species $i$ or species $j$) is the competitive dominant, and in the absence of niche differences excludes the inferior competitor. This expression shows that $\frac{\kappa_j}{\kappa_i}$ combines two fitness components, the "demographic ratio" $\left(\frac{\eta_j-1}{\eta_i-1}\right)$ and the "competitive response ratio" $\left(\sqrt{\frac{\alpha_{ij}\alpha_{ii}}{\alpha_{ji}\alpha_{jj}}}\right)$. The demographic ratio is a density independent term and describes the degree to which species $j$ has higher annual seed production, per seed lost from the seed bank due to death or germination, than species $i$

$$\eta_j = \frac{\lambda_j g_j}{1 - \left(1 - g_j\right)s_j}.$$

The competitive response ratio is a density-dependent term, which describes the degree to which species $i$ is more sensitive to both intra and interspecific competition than species $j$. Note that the same interaction coefficients defining niche differences are also involved in describing the competitive response ratio, although their arrangement is different. Because of this interdependence, a change in interaction coefficients $(\alpha'_{ji}s)$ simultaneously changes both stabilizing niche differences and average fitness differences[21].

With niche differences stabilizing coexistence and average fitness differences promoting competitive exclusion, the condition for coexistence (mutual invasibility) is expressed as[14,38]:

$$\rho < \frac{\kappa_j}{\kappa_i} < \frac{1}{\rho}. \qquad (4)$$

This condition shows that species with large differences in fitness need to also have high niche differences to coexist. In contrast, species with similar fitness can coexist even with small niche differences. As a consequence, the mutual invasibility criterion allows us to quantify the degree to which a pair of species can stably

coexist. Species pairs whose niche differences are much larger than the minimum required to overcome the fitness differences between them will be more stable than species pairs whose niche differences are close to the minimum. Species pairs whose niche differences are smaller than the minimum needed to overcome fitness differences will be unstable. We used this condition to relate the degree of stability to productivity (see below "Analyses" section).

**Field parameterization of population models under two contrasting climatic conditions**. We conducted a field experiment to parameterize these models with estimates of species germination fractions, seed survival in the soil and per germinant fecundities in the absence of neighbors. We also estimated all pairwise interaction coefficients between the species by growing each species in competition with itself and with all other species, in experimental plant communities in which we manipulated competitor density and identity, following previous experimental designs[18]. Specifically, we established 180 rectangular plots (0.65 m × 0.5 m) in September 2014 prior to the major autumn rains. We randomly assigned each of 80 plots to be sown with one of the ten species at a density of 2, 4, 8, or 16 g m$^{-2}$ of viable seed, giving two replicates per density and per species. Each plot was divided into 20 subplots (a four row by five column array) with a buffer of 2 cm along the edge of the plot. At the center of each subplot, we sowed five viable seeds of one of the ten species, and germinants were thinned to a single individual per subplot. With this experimental design, we estimated each species' germination fraction ($g_i$) by counting the number of germinants and dividing by the total number of seeds originally sown in each plot. We also measured viable seed production on two focal individuals per species and plot, when they were competing with different numbers of neighbors of the same species, and with each of the other nine species ($N_j$) within a radius of 7.5 cm. We additionally established ten plots that had the same array but did not include any density treatment in order to measure viable seed production of focal individuals of the ten species in the absence of competition. Information from plots both with and without density treatments were combined to estimate per germinant seed production in the absence of neighbors ($\lambda_i$) and the interaction coefficients ($\alpha_{ij}$) according to the function[18].

$$F_i = \frac{\lambda_i}{1 + \sum_j \alpha_{ij} N_{j,t}}. \tag{5}$$

To fit this function, we used a maximum likelihood approach (optim method = L-BFGS-B with log-norm error structure) to ensure that $\lambda_i \geq 1$ because negative germinant fecundities are not biologically meaningful. However, pairwise interaction coefficients ($\alpha_{ij}$) were not bounded to any specific range. This procedure allows us to estimate the strength of both competitive and facilitative interactions between pairs of species. For each target species $i$, we fit a separate model jointly evaluating its response to individuals of all other species and itself. This approach fits a single per germinant fecundity in the absence of competition, $\lambda_i$ for each species $i$. With this modeling approach, we found that competitive interactions were prevalent in our system. All pairwise interactions were positive (i.e., competition) under the control climate, and only two pairwise interactions were negative (i.e., facilitation), but close to zero, under the drought treatment. Although facilitation can be a source of complementarity[15], we did not consider it in our further analyses because it was so rare.

Finally, to obtain the seed bank survival ($s_i$), we followed the method detailed in[38], burying five replicates of 100 seeds each on the surrounding area from September 2014 to August 2015 and determining their viability as described in ref. [7]. Finally, we repeated the same experiment with the remaining 90 plots, sowing seeds on 10th December 2014 to simulate a drier climate. We selected this type of treatment because annual species germination only occurs after major autumn rains and, in Mediterranean ecosystems, delays in the start of the rainy season strongly affect annual plant population dynamics[41]. This delay of 64 days resulted in changes in daylight, temperature, and rainfall between treatments. However, most notably, it produced a 38.7% reduction in precipitation (from 532 in the first experiment to 326 mm for this second experiment).

**A biodiversity-functioning experiment with multiple functions**. We conducted a biodiversity-functioning experiment to simultaneously estimate complementarity and selection effects for three different functions: biomass production, litter decomposition, and changes in soil nutrient content. We established 104 circular plots (0.75 m$^2$) in the same area and at the same times as the competition experiment. We randomly assigned each plot to be a monoculture or a mixture of 3, 5, 7, 9, or 10 species. All plots were sown at a total seed density of 15 g m$^{-2}$, and seed mass was evenly divided between the species in mixtures. To create the mixtures, we randomly assembled six different communities of three species, four communities of five species, three communities of seven species, and two communities of nine species. These communities, as well as the ten monocultures and the one 10 species mixture, were all replicated twice within each climatic condition (i.e., climate control and drought). We visually assessed the biomass of each plot biweekly, and collected aboveground biomass when it was maximal in each plot. We defined the peak of biomass as the first date when a majority of species were senescent. At this time, all species had produced flowers. Biomass was separated by species, air dried for 2 weeks, then oven dried at 60 °C during 3 days and weighed (g).

In addition, we conducted biweekly surveys of leaf senescence within species to estimate when to put litter bags in the soil. During these surveys, we collected senesced leaves to fill litter bags, which were placed in the ground at the peak of leaf senescence. We defined the peak of leaf senescence as the date when the number of individuals with clear senescence symptoms (several leaves dropped from the individuals) outnumbered those without. These litter bags initially contained between 0.35 and 1.5 g of leaf litter material from a single species, which was collected from individuals of the same plot where we placed the bags. We therefore avoided pooling litter from different plots to ensure that litter quality and litter decomposition rates are driven by the specific species traits and competitive, soil, and microenvironmental conditions of each plot. We separated litter bags for each of the species included in the plot. This might underestimate litter mixing effects but the alternative, a single litter bag with mixed litter, would not have allowed us to distinguish the identity of decomposed litter and therefore to estimate decomposition rates at the species level. After 3 months, litter bags were harvested, carefully brushed clean, dried at 60 °C during 3 days, and weighed to calculate the percentage of litter mass loss.

We assessed soil nutrient dynamics as changes in C, N, P, and K, Ca, Mg cations right before (September 2014) and after the experiment (September 2015), in the first 10 cm of soil. This corresponds to the soil depth influenced by annual plant vegetation in Mediterranean ecosystems and contains 95% of the total community root biomass[42]. For chemical analyses, soils were dried in the lab at 30 °C until constant weight, and sieved (2 mm) to eliminate stones and large roots. Soils were analyzed for total organic C (%) (Walkley-Black method[43]), total organic N (%) (Kjeldahl method[44]), available P (mg/kg) (Olsen method)[45], and exchange cations (mg/kg) (Ca$^{2+}$, Mg$^{2+}$, K$^+$, extracted with 1 M ammonium acetate and determined by atomic absorption).

**Analyses**. We first explored the relationships between species diversity and biomass production, litter decomposition and soil nutrient contents at the end of the experiment. We tested for linear and nonlinear saturating relationships for the three types of functions using diversity interaction models[20].

Then, we tested for correlations between complementarity/selection effects and niche/fitness differences, under the two climatic conditions and for the three functions considered. Because niche and fitness differences are defined as pairwise measures, we could not use the standard additive partitioning approach to calculate them[10] and instead we used diversity interaction models[20] to calculate measures analogous to complementarity which can be estimated for each pair of species and measures analogous to selection effects, which can be derived for each individual species. Although diversity interaction models were not originally built to estimate selection and complementarity effects, we reinterpret them as providing measures that relate to selection and complementarity conceptually and empirically. Species selection effects were estimated as the main effect of each species on each function and large values therefore indicate species that provide high levels of function when they dominate communities. This is analogous to the selection effect, which is high and positive when functioning in mixtures is dominated by species with high monoculture functioning. Complementarity effects occur when species on average increase their functioning in mixture compared to monoculture and therefore when functioning in mixtures is delivered by multiple species. Complementarity effects therefore occur when species compete less strongly with each other and perform better in mixtures than monocultures because for instance they partition resource use or they dilute each other's specialist enemies. We therefore consider the pairwise interactions between species from the diversity interaction model to indicate complementarity between them. In order to convert selection effects to a pairwise measure we calculated the ratio between "selection effects" (intercepts from diversity interaction model) for pairs of species. We used the ratio rather than a difference because fitness differences are also defined as a ratio between species fitnesses (see Eq. 3). We then checked whether our measures of selection and complementarity effects derived from the diversity interaction models[20] correlated with the original effects produced by the additive partition of Loreau and Hector[10]. In order to do this, we summed the individual (selection), or pairwise (complementarity) values from the diversity interaction models across all species in each community. These values correlated reasonably well with the values from the additive partitioning ($r$-values ranging between 0.487 and 0.769; Supplementary Fig. 2).

We used Mantel tests, and the Benjamini and Hochberg correction for multiple comparisons, to test for significant correlations between coexistence (niche and fitness differences, Eqs. 2, 3) and biodiversity-functioning mechanisms (complementarity and selection effects). In addition to analyzing the overall fitness differences we also split them into their two components, the demographic ratio and the response to competition ratio, and correlated each component with complementarity and selection effects. The same Mantel test procedure was also used to test for the correlation between the stability of species pairs (difference between the observed niche difference and the minimum niche difference needed to allow coexistence) and the degree of function predicted for that pair. We used our diversity interaction models to estimate the degree of functioning predicted for each species pair. Finally, we analyzed whether niche differences need to exceed fitness differences to maximize function or whether large niche differences alone are sufficient to lead to high ecosystem function. We derived a metric that combines the effect of both determinants of competitive interactions following Eq. 4,

we computed for each species pair its excess of niche differences, i.e., the extent to which niche differences exceed those necessary for stable coexistence. The metric was derived as the observed niche differences minus the niche differences needed to offset the observed average fitness differences between the species. A more positive excess of niche differences means that the species pair can coexist more stably whereas a more negative value indicates the opposite. We then determined whether niche differences alone, or the excess of niche differences, correlated better with predicted functioning for the species pair by comparing the correlation coefficients between the two measures of niche differences and ecosystem functioning using package "cocor" version 1.1-3[46]. All analyses were conducted in R Version. 3.5.3[47].

**Reporting summary**. Further information on research design is available in the Nature Research Reporting Summary linked to this article.

## Data availability

Estimations of species demographic parameters, pairwise competitive coefficients, and predicted pairwise functioning for the three functions evaluated under control climate and drought conditions are publicly available at https://doi.org/10.6084/m9.figshare.12578444.

## Code availability

Code used to estimate interaction coefficients can be found at "cxr" (CoeXistenceR) open repository https://zenodo.org/record/3909328#.XvY0zZP7T_8. This R package (version 1.0.0) has been developed by *Radical Community Ecology* https://github.com/RadicalCommEcol, a Github organization integrating the labs of Oscar Godoy and Ignasi Bartomeus committed to understand the complexity of ecological communities with radical modeling tools and open science. Code used to estimate diversity interactions models (model 2A) can be found at https://besjournals.onlinelibrary.wiley.com/doi/10.1111/1365-2745.12052, Appendix S6.

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

## Acknowledgements

We thank Eduardo Gutierrez and Juan S. Cara for conducting soil nutrient analyses and lab guidance. O.G. acknowledges postdoctoral financial support provided by the European Union Horizon 2020 research and innovation program under the Marie Sklodowska-Curie grant agreement No 661118-BioFUNC.

## Author contributions

O.G., L.G.A., and E.A. designed the study. O.G., I.M.P.R., and L.M. conducted field and lab work. O.G. analyzed the data. O.G. wrote the first draft of the paper and E.A., L.G.A., I.M.P.R., and L.M. contributed substantially to revisions.

## Competing interests

The authors declare no competing interests.
