## [Peer Review File · Nature Communications]

Reviewer #1 (Remarks to the Author):

The authors present an empirical comparison of two areas of modern community ecology: coexistence theory (e.g. Chesson 2000) and BEF theory (e.g. Loreau & Hector 2001). Unfortunately, the authors seem focused on testing how these two theories compare to one another for their own sake, not because the comparison reflects an interesting mechanistic question. While unifying community ecology is important, doing it phenomenologically (instead of mechanistically) will lead us down a path of wasted time and money. I would recommend that the authors dig deeper into the mechanisms underlying both Chesson's stabilizing vs. equalizing framework and Loreau & Hector's complementarity vs. selection framework. For example, facilitation can be a stabilizing mechanism, but only when interspecific facilitation is greater than intraspecific facilitation. In either case, the α terms must be less than 0, but this is systematically prohibited in the coding of their model. The authors are familiar with the applications of these theories, but I would also recommend they consult with a mathematical ecologist to further understand the implications of the assumptions they are embedding in their models. Beyond this, a single year of data is insufficient to test the proposed theories.

Major errors/issues:

1. Introduction and framework: Complementarity effects as calculated by Loreau & Hector are not mechanistic. At least three classes of mechanisms fall under this umbrella: resource partitioning, abiotic facilitation, and pathogen pressures. Importantly, recent work has shown that resource partitioning is likely not the most important of these mechanisms (e.g. Barry et al. 2019). Furthermore, both Chesson's stabilizing mechanisms and equalizing mechanisms could include abiotic facilitation: if interspecific facilitation is greater than intraspecific facilitation, this would be stabilizing, but if microclimate amelioration homogenizes a plot and makes demographic rates more similar to each other, this would be an equalizing mechanism. The authors need to take a surgical approach to this problem in order to not further muddy an already muddy subject.

Some examples: Lines 59-61: Complementarity effects (as calculated by Loreau & Hector) are not mechanistic. Lines 66-67: Because complementarity effects are not mechanistic, this doesn't seem like the right place to start. Lines 68-73: this is all assuming that the most important mechanisms are driven by competition...but this is likely not the case. Lines 74-76: Linking these two theories to one another is not interesting a priori...

2. Methods: Overall the biggest methodological issue is that the experiment only ran for one year. Even in an annual plant community, there is no reasonable way to parameterize a model about stability in a one-year-old community. An ecological community results from more than just a single life cycle of plants (e.g. leaf litter accumulation, pathogens, soil community, etc). Plenty of work in the BEF literature demonstrated that BEF effects don't stabilize for at least several years (probably closer to 10 years).

Beyond this, the modelling approach had some issues and needed clarification in some areas.

Line 314: shared mutualisms would not result in niche differences, please clarify

Line 318: Be clear in your discussion of competitive dominance (relative values for α) vs. higher fitness values. You use them interchangeably in the text and this has serious implications for interpretation of the model

Lines 324-327: when there is perfect niche difference (species only compete with conspecifics, the competitive response ratio goes towards infinity, how is this handled?

Lines 360-361: you cannot restrict α values to be greater than 0 and compare this with complementarity effects. Complementarity effects are likely strongly driven by facilitation and facilitation in this framework is represented by negative α values.

3. Results: the lack of clear mechanistic justification for these comparisons becomes very clear in your results. Also, you overstate the results repeatedly, which makes the reader less confident in

the work in general. For example, in Figure 2, complementarity effects are ONLY clearly linked to niche differences for biomass production. For both litter decomposition and soil nitrogen, selection effects are much more clearly correlated with niche differences (while in a negative direction the correlation is still clear). This means that you have more support for a link between selection effects and niche differences. There is no mechanistic reason why that would happen and also no theoretical underpinning for why that would happen. Again, this makes the lack of theoretical underpinning for the whole paper particularly clear. Similar issues are repeated throughout Figure 2.

In Figure 3, it does look like there is an interesting pattern in control climate conditions in terms of biomass production, but then the rest of the figures make it seem like this could just as easily have been a spurious correlation. None of the rest of the relationships support the proposed hypothesis at all.

Reviewer #2 (Remarks to the Author):

The manuscript by Godoy and colleagues is a novel attempt to combine predictions and insights from coexistence theory to understand complementarity and selection effects for biodiversity-ecosystem function studies. This work uses an elegant experiment to parameterize two sets of models and shows how complementarity/selection are related to niche and fitness differences. This is a much-needed experiment and an extremely valuable contribution to the literature. I really like this study, though I do have some suggestions for improvement.

The environmental treatment seemed to come out of nowhere in the introduction but is actually a cool feature of the experiment. That you can create different scenarios of niche and fitness differences with the same species is a powerful way to test some of the central questions. The importance of environmental differences for influencing both sets of mechanisms (niche and fitness differences and complementarity/select) should be introduced earlier.

The final analyses looking at competitive networks and equilibrium coexistence, to me, seem underdeveloped and the main finding, that the full community will not coexist and thus diversity and function are transient, do not come across as robust. My general concern about niche/fitness difference experiments is that they are temporally and spatially limited and that parameter estimates are biased by local conditions without accounting for coexistence mechanisms influenced by spatial or temporal niche differences. None of these types of experiments has been able to fully account for full community coexistence, and I believe that the methodology is too limited. Thus, I recommend removing this test (lines 150-156, 221-229), as, to me, it doesn't really test the main hypothesis anyway.

In general, I found that the discussion lacked biology, as opposed to ecological theory. I think it would be helpful to bring the results back to the system and species manipulated and to put the mechanisms into terms that are germane about this system. A little more about species and those contributing to selection effects and the different ecosystem functions for example. In another example, the section on the environmental variability again lacks biology from the species, which least sensitive and how did they contribute to selection effects and functioning. And again, like the introduction, the environment treatments seems disconnected from the other results, rather than being a way to strengthen the findings.

Abstract -second sentence seems too obtuse. Please be more precise about mechanisms/classes. And the third sentence - what the prediction refers to is not clear.

Line 67-68: not clear what 'its effects' is referring to exactly.

Line 72: 'may' to 'might'

Line 73: maybe give a mechanistic explanation or example of how this could happen.

Lines 77-89: great paragraph explaining your questions and motivation.

Lines 95-96: Seems a little obtuse, please be more precise on how Environmental variation influences the link between diversity and productivity.

Lines 127-130: Even though described in methods, I think a little more description of what was meant by demographic and competitive response ratios and how these were quantified.

Lines 142-145: you state twice 'species pairs predicted to stably coexist', which is a mouthful. Why stably, is it necessary, just 'predicted to coexist' should be sufficient.

Line 161: "Both fields' -I know what you mean, but you haven't really said what these are in the previous sentence.

Lines 207-219: I'm sure you can say a little more about the likely mechanisms. Which species maximized these and what is known about them?

Reviewer #3 (Remarks to the Author):

I have read "A mechanistic path to maximize biomass production while maintaining species diversity" by Godoy et al. In this contribution, the authors put to the test a number of ideas that have received recent theoretical attention on the relationship between coexistence mechanisms and diversity effects of ecosystem functioning. In this sense, the contribution is timely and important. The experiment is well designed, and the methods seem correct (although there is a critical issue that is not clearly explained, making it impossible to provide a complete assessment in this regard), and the statistical methods are correct given the data. I have some doubts on the hypothesis that ecosystem functioning increases when niche differences overcome fitness differences, allowing for stable coexistence, rather than increasing as a result of adding the effects of niche and fitness differences considered separately. There is no clear reasoning behind this idea, previous theoretical work does not support it (and rather contradicts it), and the results do not support it clearly. However, this is not a critical issue, and it does not reduce the merit of the contribution. There is also an issue on a possible extrapolation that could cast doubts on one important result. Below I list my comments as issues appear in the manuscript. The ones with an asterisk are the ones that I believe are most important.

1. Title: when I first read the title I thought that a path was being proposed to explain some kind of paradox precluding coexistence in productive systems. I would prefer something in the lines of "Coexistence stabilization has a positive effect on ecosystem functioning in diverse annual-plant communities".

2. Abstract: The problem is posed as "it is important to know how these two things relate to each other". I would prefer to see an argument for why they should relate at all (especially if we are told in the abstract that theory states that they do not "map to each other"). Fitness differences (sensu Chesson), complementarity and selection (sensu diversity-functioning theory) are not widely used terms that may require a definition in the abstract. I can see that this is a challenge given 150 words. Please conduct some copyediting for grammar.

3. L31: Plants or plant species?

4. L63: Isn't it the other way around? Niche differences strengthen stabilization.

5. L82: large niche differences promote evenness. Yes, in relative terms when compared to a very specific community (one with less differentiation, *ceteris paribus*; see Turnbull et al 2013). In general, there is no reason to expect evenness in absolute terms if niches are different. Population sizes would depend on the availability of the resources used by the different species, which is an environmental factor independent of niche differentiation. The authors have tried to be very concise in the presentation of their hypotheses and predictions, but I feel that more space is needed to explain them clearly. Please see comments 6 and 7.

6. L82: You have not argued for evenness before as a factor that increases functioning. The function-evenness relationship needs to be clearly stated before (Turnbull et al 2013, Connolly et al 2013 may be cited).

7. *General hypothesis: Perhaps I am not getting something right, but I am not convinced about the way the argument presented. In L82-84, niche and fitness differentiation are discussed as

having somehow independent effects on functioning: function increases with niche differences (ND) and decreases with fitness differences (FD.) The expected pattern (greater function as ND increases relative FD) would be expected if the "independent" effects of ND and FD are simply added, but also if it is the difference ND-FD that matters. However, you favor the latter alternative. Please state more clearly the biology behind this argument.

8. *L84: (In line with previous comment) "sufficient niche differentiation to offset fitness differences". This is a binary variable (sufficient vs insufficient), while I believe that, from the argumentation in L 82-84, it follows that a greater difference between niche differences (ND) and fitness differences (FD) should result in a greater functioning: no digital divide expected, gradual change instead. See also Turnbull et al. 2013. They show that overyielding is related to ND, regardless of whether ND is sufficient to grant stable coexistence (i.e., no need for $ND > FD$). In fact, overyielding does not provide evidence on stable coexistence at all.

9. L123: The way this is written suggests that p value should be used as effect size. In any case, p values are not as important as effect sizes in this context. Please rewrite.

10. *L141-148, Fig. 3A-B. You would expect to see more biomass towards the bottom-right corner of the graphs if the effects of ND and FD just add up or if yield only increases when there "sufficient niche differentiation to offset fitness differences" (L84). Therefore, I do not see any reason to conclude in favor of the second alternative. A more sophisticated analysis would be required to tell between both hypotheses because their predictions are so similar (gradual increase towards the coexistence region of the graphs vs. a sharp increase at the divide between coexistence and exclusion). The pattern seems quite noisy to provide a clear cut between both patterns.

11. L359: (20)?

12. Equation after L359 (and elsewhere): When the experiment was set up, seeds of the focal and interacting species were added in different amounts measured as mass of seeds (2-16 g m⁻²), but the fitted model requires numbers (not mass) of individuals. Did you count the number of interacting individuals that got established in each plot, and used that number to fit your models? Please explain.

13. L360-61. I guess that the parentheses should close after "greater or equal to zero" rather than after "BFGS-B").

14. L363: g_i was calculated as the fraction of seeds that germinated out of the five that were sown in each plot, from the number of seeds added when setting the density treatment (those added as grams per m), or both?

15. L389 and ss: I did not follow clearly how was litter manipulated and decomposition measured.

16. *L416 and ss: I like the idea of resorting to a procedure such as that proposed by Connolly et al. 2013 to solve the problem, and the fact that the procedure proposed here correlates with the "canonic" procedure of Loreau and Hector (although sometimes the correlation is quite weak, something that merits some comment from the authors) suggests that the Godoy et al.'s method used is appropriate. However, much more detail is required here. Connolly et al.'s procedure needs to be explained for the reader, and, very importantly, the way in which it was adapted/interpreted in terms of complementarity and selection needs to be thoroughly explained. In eq. 1 in Connolly's paper, the model for no diversity effects is explained. I can but guess that the model used by Godoy et al is eq. 2b from Connolly, which is the same as eq 1 with only one more term comprising diversity effects ($\Delta \pi_j$ to the power θ). If I get it right, positive Δ s result in overyielding and thus indicate complementarity, and θ is a shape parameter. I can't see where the measurements of selection come from (unless θ can be interpreted in this way, but it is not obvious why). These things need to be explained, especially because Connolly et al. do not propose any interpretation of their model as a partition of diversity effects into complementarity and selection effects and thus Godoy et al.'s proposal seems to be entirely novel.

17. L421: "because it possesses"

18. L428: Please provide a short explanation of what Loreau and Hector do, and why the results of your procedure needs to be compared with those derived from their method.

19. L447: delete "this means that a"

20. L451: This is probably a personal bias, but I find the term "network analysis" a bit misleading here. What you are using here has long been known simply as stability analysis. Network analysis suggests to me the study of the structure of the matrix such as compartmentalization, connectivity, nestedness, small-world structure etc.

21. Fig 1 "Non-linear instead regressions fitted the data better" remove "instead" and: better than what?

22. *Fig 3: If there are no species pairs in the coexistence region under drought conditions (Figure S3), the complementarity effect observed in the coexistence region in panel B may be a strong extrapolation and thus may not be trusted. This would cast doubts on some important conclusions of the manuscript. Am I missing something?

23. Fig S1: I am a bit surprised by the fact that the relationships between diversity and functioning are straight lines. Connolly et al model corresponds to a not linear relationship. What model are you using here?

24. The caption of figure S5 states "see Figure S5", which is odd. Is this an error and you want the reader to see some other figure instead? Besides, I find the figure unclear.

Responses to reviewers' comments.

Reviewer #1

COMMENT 1: The authors present an empirical comparison of two areas of modern community ecology: coexistence theory (e.g. Chesson 2000) and BEF theory (e.g. Loreau & Hector 2001). Unfortunately, the authors seem focused on testing how these two theories compare to one another for their own sake, not because the comparison reflects an interesting mechanistic question. While unifying community ecology is important, doing it phenomenologically (instead of mechanistically) will lead us down a path of wasted time and money. I would recommend that the authors dig deeper into the mechanisms underlying both Chesson's stabilizing vs. equalizing framework and Loreau & Hector's complementarity vs. selection framework. For example, facilitation can be a stabilizing mechanism, but only when interspecific facilitation is greater than intraspecific facilitation. In either case, the α terms must be less than 0, but this is systematically prohibited in the coding of their model. The authors are familiar with the applications of these theories, but I would also recommend they consult with a mathematical ecologist to further understand the implications of the assumptions they are embedding in their models. Beyond this, a single year of data is insufficient to test the proposed theories.

RESPONSE: We thank the reviewer for their overall view of the comparison between coexistence and biodiversity functioning mechanisms. We do feel that it is interesting to attempt to unify the main groups of coexistence and biodiversity-functioning mechanisms. Both fields have been successful in uniting the many hundreds of individual mechanisms that can drive coexistence or effects of biodiversity on function into two main classes and we feel that showing how these classes of mechanism relate to each other will give us a more unified view of how changes in coexistence mechanisms alter functioning. It would of course also be interesting to try to link all the individual coexistence and diversity-functioning mechanisms (e.g. partitioning of different resources, actions of multiple natural enemies, spatial and temporal heterogeneity, facilitation, etc., etc.) but this will be a huge task given the number of mechanisms involved and we feel that starting with the main classes of mechanism allows us to make progress in unifying the fields and ask basic questions like: do stably coexisting communities have higher functioning than communities where the species do not stably coexist? In this study we have therefore tested for an empirical relationship between biodiversity and coexistence mechanisms, showing for the first time that they have a complex interrelationship. This contrasts with some early assumptions that niche differences should link only to complementarity effects and differences in fitness should be associated with differences in selection effects. Most importantly we show that the most stably coexisting communities are those with highest function, but only for biomass. This shows that the conditions for stable coexistence relate to the conditions for high biomass production, which is an important link between the two bodies of research and establishes a solid connection between the drivers promoting the maintenance and functioning of ecological communities. We have added some more explanation of why we feel it is interesting to link overall processes to the introduction.

With respect to facilitation, it is true that it can be a mechanism driving higher ecosystem functioning in diverse communities and could be one of the underlying mechanisms driving complementarity effects (as proposed by Wright et al. 2017). However, there are only a few empirical studies explicitly showing that facilitation drives complementarity and ecosystem functioning (e.g. Cardinale et al 2002). Even Cardinale et al. 2002 does not present estimates of the per capita effect of facilitation of one species on another, and only considers interspecific facilitation. In fact, very few biodiversity experiments compare species performance in mixture with the performance of individuals alone, without competitors, which would be necessary to demonstrate that facilitation is operating. Most comparisons are between monoculture and mixture performance, which cannot say anything about facilitation. Chesson's (2013) definition of niche differences does not consider facilitation, however they can be incorporated into modern coexistence theory. To test whether facilitation does play any role in driving coexistence or effects of biodiversity on function, we have redone our analyses. We estimated

competition coefficients using algorithms that do not constrain the alpha values, so they can also be negative (please note that a negative coefficient in a Beverton-Holt function as presented in our equations means facilitation). We find that only two interspecific pairs out of 90 show weak facilitation, however, it is close to zero and therefore probably indicates no interaction rather than facilitation (see response to comment 10 for specific facilitation values). The range of competitive interactions (alpha values) goes from 0.07 to 4.13. Because of these weak positive interactions coefficients, we do not discuss facilitation in detail, given its rarity in our system.

Finally, we fully agree that including more years (and sites) is always valuable in ecological research. However, our experiment can be seen as a proof of concept showing how the field of species coexistence and ecosystem functioning can be interrelated. Performing the experiment across more years will mostly result in variation in niche and fitness differences due to environmental variability. However, it would be extremely challenging to maintain the experiment with annuals, as we would have to collect seed and resow species. We also feel that given this is an annual system running the experiment for multiple years is less critical than in perennial grassland experiments with long lived individuals (like most terrestrial biodiversity experiments). Further, we do consider different environmental conditions in the comparison between control climate conditions and delay of rainfall accounts.

References:

- Bulleri, F., Bruno, J. F., Silliman, B. R., & Stachowicz, J. J. (2016). Facilitation and the niche: implications for coexistence, range shifts and ecosystem functioning. *Functional Ecology*, 30(1), 70-78.
- Cardinale, B. J., Palmer, M. A., & Collins, S. L. (2002). Species diversity enhances ecosystem functioning through interspecific facilitation. *Nature*, 415(6870), 426-429.
- Chesson, P. (2013). Species competition and predation. In *Ecological systems* (pp. 223-256). Springer, New York, NY.
- Wright, A. J., Wardle, D. A., Callaway, R., & Gaxiola, A. (2017). The overlooked role of facilitation in biodiversity experiments. *Trends in Ecology & Evolution*, 32(5), 383-390.

COMMENT 2: Introduction and framework: Complementarity effects as calculated by Loreau & Hector are not mechanistic. At least three classes of mechanisms fall under this umbrella: resource partitioning, abiotic facilitation, and pathogen pressures. Importantly, recent work has shown that resource partitioning is likely not the most important of these mechanisms (e.g. Barry et al. 2019). Furthermore, both Chesson's stabilizing mechanisms and equalizing mechanisms could include abiotic facilitation: if interspecific facilitation is greater than intraspecific facilitation, this would be stabilizing, but if microclimate amelioration homogenizes a plot and makes demographic rates more similar to each other, this would be an equalizing mechanism. The authors need to take a surgical approach to this problem in order to not further muddy an already muddy subject.

RESPONSE: We thank the reviewer for pointing out this reference and we fully agree that there are many underlying mechanisms that can drive complementarity and selection effects, niche and fitness differences. We also agree that resource complementarity may not be the main driver of complementarity effects and we have cited the review paper by Barry et al. to explain that many processes can influence complementarity effects and have checked that we make this clear throughout the paper. Because of the complexity of these multiple underlying mechanisms, we think it is interesting to distinguish whether effects of diversity on functioning are driven by many species (complementarity) or rather by a few (selection), which is the main point of the additive partition, and to determine how this links to the degree of niche and fitness differences.

As with complementarity effects, niche differences can be influenced by resource partitioning, multitrophic interactions, and natural enemies. We do not however agree with the reviewer statement that Chesson's stabilizing mechanisms can include facilitation. The case the reviewer is explaining is a very particular case of facilitation in which species within the community experience both intra- and inter-specific facilitation. Unfortunately, we are not

aware of any study in the literature that has shown this pattern. In most cases, facilitation occurs between species but not between and within species at the same time. The reviewer is right that in Chesson's 2013 definition of niche differences, facilitation could be included if all alpha terms are negative. Otherwise niche differences cannot be estimated because of the square root term of the equation (Eq. 1). As we said previously, when P. Chesson presented his definition for the first time, he did explicitly mention how niche differences vary with competition and predation but not with facilitation.

$$\rho = \sqrt{\frac{\alpha_{ij} \alpha_{ji}}{\alpha_{jj} \alpha_{ii}}} \quad \text{Equation 1.}$$

The other mechanisms that the reviewer comments on driving equalizing mechanisms is what we found in our drought treatment. This is captured by one of the two components of the species fitness. The demographic component (the left component of the fitness differences ratio κ_j/κ_i) that includes all species vital rates (seed production in the absence of competition, germination rates and soils survival rates in the soil bank summarised in the eta term (Eq. 2) (see lines 346-354 for detailed explanation). Therefore, this abiotic effect of species coexistence on equalising effects and on ecosystem functioning is well-characterised in our system.

$$\frac{\kappa_j}{\kappa_i} = \left(\frac{\eta_j - 1}{\eta_i - 1} \right) \sqrt{\frac{\alpha_{ij} \alpha_{ii}}{\alpha_{jj} \alpha_{ji}}} \quad \text{Equation 2.}$$

COMMENT 3: Some examples: Lines 59-61: Complementarity effects (as calculated by Loreau & Hector) are not mechanistic. Lines 66-67: Because complementarity effects are not mechanistic, this doesn't seem like the right place to start. Lines 68-73: this is all assuming that the most important mechanisms are driven by competition...but this is likely not the case. Lines 74-76: Linking these two theories to one another is not interesting a priori...

RESPONSE: Defining what is a mechanism is complex (and a large and active area of philosophy) and mechanisms can form hierarchies so that higher level mechanisms (complementarity effects) can be decomposed into lower level mechanisms (like species partitioning forms of N or specialist soil pathogens reducing monoculture performance). We would also note that complementarity and selection are often referred to as "classes of mechanisms", e.g. in the original Loreau and Hector (2001) study. However, in order to avoid confusion and because the term is somewhat controversial, we have rewritten the introduction to avoid the term "mechanistic" or "mechanism". We have also checked that we do not imply that complementarity or niche differences are largely driven by resource competition and we have explained why we do think it is interesting to link these bodies of theory, see also comment 1.

COMMENT 4: Methods: Overall the biggest methodological issue is that the experiment only ran for one year. Even in an annual plant community, there is no reasonable way to parameterize a model about stability in a one-year-old community. An ecological community results from more than just a single life cycle of plants (e.g. leaf litter accumulation, pathogens, soil community, etc). Plenty of work in the BEF literature demonstrated that BEF effects don't stabilize for at least several years (probably closer to 10 years).

RESPONSE: The reviewer is right that many biodiversity ecosystem functioning (BEF) experiments have run for several years and that experiments in perennial grasslands do often show that biodiversity effects change over time as species shift their abundances and soil conditions change. However, in an annual community it is perhaps less likely that there will be such long-term changes in biodiversity-functioning relationships given that the communities must recruit from seed each year (rather than infrequently in a perennial system with long lived individuals). We chose to work in an annual system because it is much easier to model annual plant population dynamics, however, methods are being developed to extend our approach to perennial systems and it will be interesting to estimate how these relations change over time. In

annual systems, we would expect that biodiversity-functioning relationships might fluctuate over time, rather than changing directionally, due to changing weather conditions and we do address the importance of altered environmental conditions. We include two treatments (control climate and drought) and we show that these do create important variation in niche and fitness differences and in complementarity and selection effects across three different functions (biomass, litter decomposition, and nutrient accumulation). We therefore do incorporate environmental variability and test whether our results hold across altered environmental conditions. Finally, the reviewer says that one year of data is not enough to evaluate stability conditions in annual plant communities. The reviewer is completely right and more years of data would be ideal to get better estimates of competition coefficients, however, coexistence studies very rarely incorporate multiple years of data. It is also important to mention that that we are not focused on assessing stability, rather we are focused instead on the two mechanisms of MCT, niche and fitness differences and how the variation of these two species differences across environmental conditions are linked with effects of biodiversity on functioning.

COMMENT 5: Results: the lack of clear mechanistic justification for these comparisons becomes very clear in your results. Also, you overstate the results repeatedly, which makes the reader less confident in the work in general. For example, in Figure 2, complementarity effects are ONLY clearly linked to niche differences for biomass production. For both litter decomposition and soil nitrogen, selection effects are much more clearly correlated with niche differences (while in a negative direction the correlation is still clear). This means that you have more support for a link between selection effects and niche differences. There is no mechanistic reason why that would happen and also no theoretical underpinning for why that would happen. Again, this makes the lack of theoretical underpinning for the whole paper particularly clear. Similar issues are repeated throughout Figure 2.

RESPONSE: We thank the reviewer for pointing out this lack of clarification. We already said in the original submission that our main finding was that ecosystem functioning related to stable coexistence, only for biomass production (see lines in results 139-149). In fact, we had a whole paragraph in the discussion arguing why we did not find the relationship between stable coexistence and high functioning for litter decomposition and nitrogen cycling (lines 263-275). We consider this to be an important and interesting result because it shows that the conditions for stable coexistence will also maximise functioning, but only if we consider a function directly linked to plant performance like biomass. If we consider other functions, then stable coexistence will not necessarily link to high functioning, meaning that we may need to consider other trophic groups to explain functioning. In terms of selection effects linking to niche differences, we feel that there is a clear expectation for a negative correlation between niche differences and selection effects. Negative selection effects arise when species which perform poorly in monoculture grow better in mixtures, i.e. for these species intraspecific competition is much stronger than interspecific competition, which is exactly the signature of stabilising niche differences. This relationship has also been shown by previous theoretical work linking niche and fitness differences to selection and complementarity (Turnbull et al. 2013).

References:

Turnbull, L. A., Levine, J. M., Loreau, M., & Hector, A. (2013). Coexistence, niches and biodiversity effects on ecosystem functioning. *Ecology letters*, 16, 116-127.

COMMENT 6: In Figure 3, it does look like there is an interesting pattern in control climate conditions in terms of biomass production, but then the rest of the figures make it seem like this could just as easily have been a spurious correlation. None of the rest of the relationships support the proposed hypothesis at all.

RESPONSE: This is exactly what we present in our discussion section, there is a significant connection between coexistence mechanisms and functioning for biomass production, but these relationships are no longer significant when we introduce functions that

involve the action of other trophic levels, such litter decomposition which also depends strongly on microbial activity. In order to demonstrate that this is not a spurious correlation, we have undertaken further modelling to test what predicts high functioning, see also response to reviewer 3.

COMMENT 7: Line 314: shared mutualisms would not result in niche differences, please clarify

RESPONSE: We respectfully disagree with this opinion. There are perspectives (Pauw 2013) and empirical work showing that shared mutualists can drive niche differences (Lanuza et al. 2018)

References:

Pauw, A. (2013). Can pollination niches facilitate plant coexistence?. *Trends in ecology & evolution*, 28(1), 30-37.

Lanuza, J. B., Bartomeus, I., & Godoy, O. (2018). Opposing effects of floral visitors and soil conditions on the determinants of competitive outcomes maintain species diversity in heterogeneous landscapes. *Ecology letters*, 21(6), 865-874.

COMMENT 8:Line 318: Be clear in your discussion of competitive dominance (relative values for alpha) vs. higher fitness values. You use them interchangeably in the text and this has serious implications for interpretation of the model

RESPONSE: We apologise for this confusion. We now make this distinction clear.

COMMENT 9:Lines 324-327: when there is perfect niche difference (species only compete with conspecifics, the competitive response ratio goes towards infinity, how is this handled?)

RESPONSE: The result that the reviewer is referring to is a mathematical possibility but this is something we did not observe in our system. There is not a single species pair where both interspecific interactions are zero, while intraspecific interactions were positive, regardless of their magnitude. Therefore, although this comment is important from a mathematical point of view it is not critical to analysing our data.

COMMENT 10:Lines 360-361: you cannot restrict alpha values to be greater than 0 and compare this with complementarity effects. Complementarity effects are likely strongly driven by facilitation and facilitation in this framework is represented by negative alpha values.

RESPONSE: We thank the reviewer for this comment. It is an important comment because we have redone all our analyses based on a new estimation of alphas that do not constrain values to be greater than 0. Specifically, we have used the algorithm in optim function implemented in R that is an implementation of the Nelder and Mead (1965) approach. It is very robust, although a bit slow, but it serves our purposes. With this new estimation of interaction coefficients, we have found no facilitation effects under control climatic conditions. Moreover, under drought conditions, only 2 interspecific coefficients out of 45 show negative values (i.e. negative value means facilitation in a Beverton-Holt models as it is the annual plant model) and the strength of these facilitation effects are close to zero. (-0.008 effect of *Calendula arvensis* on *Diplotaxis erucoides*, and -0.001 effect of *Papaver rhoeas* on *Calendula arvensis*) The range of values of competition goes from 0.04 to 3.82.

This shows that facilitation cannot drive complementarity effects in our system. In other systems this may be different and it will be interesting to extend our approach to cases where we might expect more facilitation, such as dryland or alpine environments.

References:

Nelder, J. A. and Mead, R. (1965). A simplex algorithm for function minimization. *Computer Journal*, 7, 308--313. 10.1093/comjnl/7.4.308.

Reviewer #2

COMMENT 1: The manuscript by Godoy and colleagues is a novel attempt to combine

predictions and insights from coexistence theory to understand complementarity and selection effects for biodiversity-ecosystem function studies. This work uses an elegant experiment to parameterize two sets of models and shows how complementarity/selection are related to niche and fitness differences. This is a much-needed experiment and an extremely valuable contribution to the literature. I really like this study, though I do have some suggestions for improvement.

RESPONSE: We thank the reviewer for these positive comments. We are happy to read that the reviewer thinks our approach is interesting because there is a much-needed experimental evidence of the relationship between the drivers of species coexistence and functioning.

COMMENT 2: The environmental treatment seemed to come out of nowhere in the introduction but is actually a cool feature of the experiment. That you can create different scenarios of niche and fitness differences with the same species is a powerful way to test some of the central questions. The importance of environmental differences for influencing both sets of mechanisms (niche and fitness differences and complementarity/select) should be introduced earlier.

RESPONSE: Following this comment from reviewer 2, we have introduced the experiment earlier in the introduction and mentioned the role of environmental variation in promoting variation in both niche and fitness differences as well as variation in complementarity and selection effects (see lines 92-99).

COMMENT 3: The final analyses looking at competitive networks and equilibrium coexistence, to me, seem underdeveloped and the main finding, that the full community will not coexist and thus diversity and function are transient, do not come across as robust. My general concern about niche/fitness difference experiments is that they are temporally and spatially limited and that parameter estimates are biased by local conditions without accounting for coexistence mechanisms influenced by spatial or temporal niche differences. None of these types of experiments has been able to fully account for full community coexistence, and I believe that the methodology is too limited. Thus, I recommend removing this test (lines 150-156, 221-229), as, to me, it doesn't really test the main hypothesis anyway.

RESPONSE: This is definitely true and we thank the reviewer for this comment. The competitive network analysis has now been removed.

COMMENT 4: In general, I found that the discussion lacked biology, as opposed to ecological theory. I think it would be helpful to bring the results back to the system and species manipulated and to put the mechanisms into terms that are germane about this system. A little more about species and those contributing to selection effects and the different ecosystem functions for example. In another example, the section on the environmental variability again lacks biology from the species, which least sensitive and how did they contribute to selection effects and functioning. And again, like the introduction, the environment treatments seem disconnected from the other results, rather than being a way to strengthen the findings.

RESPONSE: We thank the reviewer for this comment. In both the introduction and the discussion, we have included detailed explanations of the species biology used for the experiment. We have also discussed in lines 229-235, which functional traits of our studied species might be contributing to both niche differences and competitive response differences. These are the two components of species coexistence affected by density-dependent effects.

Abstract -second sentence seems too obtuse. Please be more precise about mechanisms/classes. And the third sentence - what the prediction refers to is not clear.

We have changed both the second and the third sentence
Line 67-68: not clear what 'its effects' is referring to exactly.

These effects were referring to selection and complementarity effects. We have rewritten the sentence to make it clearer.

Line 72: 'may' to 'might'

Done

Line 73: maybe give a mechanistic explanation or example of how this could happen.

We have now explained in detail why the combination of niche and fitness differences drive functioning (see lines 76-80).

Lines 77-89: great paragraph explaining your questions and motivation.

We thank the reviewer for this comment.

Lines 95-96: Seems a little obtuse, please be more precise on how Environmental variation influences the link between diversity and productivity.

In lines 90 to 97 we now explain clearer why we believe environmental variation change the relationship between diversity and productivity via modifications of niche and fitness differences. We basically believe that environmental modifications, in particular, drought reduces species fitness and the possibility of species to differentiate in their niches. The result of these modifications is in turn a reduction of ecosystem functioning via reduction of complementarity and selection effects.

We have explained with link in more detail. It now reads as

Lines 127-130: Even though described in methods, I think a little more description of what was meant by demographic and competitive response ratios and how these were quantified.

We have explained both ratios now in the results in lines 127-130

Lines 142-145: you state twice 'species pairs predicted to stably coexist', which is a mouthful. Why stably, is it necessary, just 'predicted to coexist' should be sufficient.

This is a good comment, we have changed the text accordingly.

Line 161: "Both fields" -I know what you mean, but you haven't really said what these are in the previous sentence.

We have clarified this sentence following reviewer's comment.

Lines 207-219: I'm sure you can say a little more about the likely mechanisms. Which species maximized these and what is known about them?

We have included an example of a species pair predicted to produce high functioning under high stable conditions in both climate control and drought conditions. See lines 212-219. These particular examples help to bring the particularities of our system explicitly in the discussion and the reviewer asked in the previous comment.

Reviewer #3

COMMENT 1: I have read "A mechanistic path to maximize biomass production while maintaining species diversity" by Godoy et al. In this contribution, the authors put to the test a number of ideas that have received recent theoretical attention on the relationship between coexistence mechanisms and diversity effects of ecosystem functioning. In this sense, the contribution is timely and important. The experiment is well designed, and the methods seem correct (although there is a critical issue that is not clearly explained, making it impossible to provide a complete assessment in this regard), and the statistical methods are correct given the data. I have some doubts on the hypothesis that ecosystem functioning increases when niche differences overcome fitness differences, allowing for stable coexistence, rather than increasing as a result of adding the effects of niche and fitness differences considered separately. There is no clear reasoning behind this idea, previous theoretical work does not support it (and rather contradicts it), and the results do not support it clearly. However, this is not a critical issue, and it does not reduce the merit of the contribution. There is also an issue on a possible extrapolation that could cast doubts on one important result. Below I list my comments as issues appear in the manuscript. The ones with an asterisk are the ones that I believe are most important.

RESPONSE: We thank the reviewer for these positive comments and for highlighting the contribution of our study. We have added an analysis to test whether it is indeed the difference between niche and fitness differences that explains functioning or the niche differences alone. We have also explained in lines 77-101 why we think biomass production

should be maximised when both niche and fitness differences are combined together. Our new analyses do suggest that it is the excess of niche differences that is important, i.e. the extent to which niche differences exceed the minimum necessary for coexistence, see below. We feel that this finding is an important novel contribution and we are grateful to the reviewer for suggesting we develop this idea and test it robustly. We also discuss how a combination of niche and fitness differences is important and that the two determinants of competitive outcomes are interrelated through density dependent effects (Song et al. 2019). We also acknowledge that the relationship between the conditions for stable coexistence and high functioning is mostly found for biomass, which we feel is an interesting point.

References:

Song, C. et al. On the consequences of the interdependence of stabilizing and equalizing mechanisms. *Am. Nat.* **194**, 627-639 (2019).

COMMENT 2: Title: when I first read the title I thought that a path was being proposed to explain some kind of paradox precluding coexistence in productive systems. I would prefer something in the lines of “Coexistence stabilization has a positive effect on ecosystem functioning in diverse annual-plant communities”.

RESPONSE: We apologise for the confusion. We have changed the title to the following: “Ecosystem functioning is maximized when niche differences are larger than required for stable coexistence”. We have avoided the term annual plant communities as we think this study is not about the particularities of our system. We rather use annual plant species as a model species.

COMMENT 3: Abstract: The problem is posed as “it is important to know how these two things relate to each other”. I would prefer to see an argument for why they should relate at all (especially if we are told in the abstract that theory states that they do not “map to each other”). Fitness differences (sensu Chesson), complementarity and selection (sensu diversity-functioning theory) are not widely used terms that may require a definition in the abstract. I can see that this is a challenge given 150 words. Please conduct some copyediting for grammar.

RESPONSE: We also thank the reviewer for this comment. We have changed the first three sentences in the abstract. Despite the abstract limit, we explain now that ecologists have long invoked coexistence mechanism to explain community overyielding in mixtures compared to monocultures. Yet, empirical test of this prediction is lacking. We have also highlighted the role of environmental variation. See lines 29-31

COMMENT 4: L31: Plants or plant species?

RESPONSE: Changed to annual plant species. Line 29

COMMENT 5: L63: Isn't it the other way around? Niche differences strengthen stabilization.

RESPONSE: We have changed this sentence as the reviewer requested to avoid confusion.

COMMENT 6: L82: large niche differences promote evenness. Yes, in relative terms when compared to a very specific community (one with less differentiation, ceteris paribus; see Turnbull et al 2013). In general, there is no reason to expect evenness in absolute terms if niches are different. Population sizes would depend on the availability of the resources used by the different species, which is an environmental factor independent of niche differentiation. The authors have tried to be very concise in the presentation of their hypotheses and predictions, but I feel that more space is needed to explain them clearly. Please see comments 6 and 7.

RESPONSE: This comment has been answered in comment 8.

COMMENT 7: L82: You have not argued for evenness before as a factor that increases

functioning. The function-evenness relationship needs to be clearly stated before (Turnbull et al 2013, Connolly et al 2013 may be cited).

RESPONSE: We have now made clear the relationship between functioning and evenness. However, we have put more emphasis on explaining this part by presenting the role of density dependent effects of modifying both niche and fitness differences.

COMMENT 8: *General hypothesis: Perhaps I am not getting something right, but I am not convinced about the way the argument presented. In L82-84, niche and fitness differentiation are discussed as having somehow independent effects on functioning: function increases with niche differences (ND) and decreases with fitness differences (FD.) The expected pattern (greater function as ND increases relative FD) would be expected if the “independent” effects of ND and FD are simply added, but also if it is the difference ND-FD that matters. However, you favor the latter alternative. Please state more clearly the biology behind this argument.

RESPONSE: This is a critical aspect of the manuscript and we have devoted more effort to explaining our main hypothesis clearly in the introduction. See lines 77-101 and we have conducted an additional analysis to test whether it is the difference between ND and FD that matters, or just the overall size of the NDs. To test whether niche differences have to exceed fitness differences to maximize functioning, or whether they just have to be high, we derived a metric that combines the niche and fitness differences. Following equation 4, we computed the excess of niche differences for a species pair over what is required for stable coexistence as follows: observed niche differences minus predicted niche differences needed to offset observed average fitness differences. A more positive excess of niche differences means that the species pair coexists more stably whereas a more negative value indicates the opposite. With these estimations, we then determined whether niche differences alone or the excess of niche differences correlated better with predicted functioning by statistically comparing correlation coefficients from both sources of niche differences using package “cocor”. This new analysis is now presented in lines 514-533. They show that it is the excess of niche differences for a species pair that promotes functioning, as the differences between niche differences and the minimum necessary for coexistence correlates with biomass production but the raw niche differences do not significantly correlate with the biomass produced by a species pair. It is also important to note that modern coexistence theory (MCT) following Chesson, originally treated niche and fitness differences as independent axes of variation, but in fact they are not independent.

References:

Song, C. et al. On the consequences of the interdependence of stabilizing and equalizing mechanisms. *Am. Nat.* **194**, 627-639 (2019).

Diedenhofen, B & Musch, J. cocor: A comprehensive solution for the statistical comparison of correlations. *PLoS one* **10**, 4 (2015).

COMMENT 9: *L84: (In line with previous comment) “sufficient niche differentiation to offset fitness differences”. This is a binary variable (sufficient vs insufficient), while I believe that, from the argumentation in L 82-84, it follows that a greater difference between niche differences (ND) and fitness differences (FD) should result in a greater functioning: no digital divide expected, gradual change instead. See also Turnbull et al. 2013. They show that overyielding is related to ND, regardless of whether ND is sufficient to grant stable coexistence (i.e., no need for ND>FD). In fact, overyielding does not provide evidence on stable coexistence at all.

RESPONSE: We have rewritten this sentence to avoid the dichotomy between sufficient versus insufficient because we believe is a continuous axis of variation. We measured such continuous axes of variation by estimating the excess of niche differences compared to the minimum necessary (see our response to previous comment number 8). Interestingly it appear that niche differences should not just be sufficient to stabilise coexistence but that an excess of

niche differences over the minimum necessary promotes functioning. Equally high niche differences alone do not promote functioning unless they exceed the fitness differences.

COMMENT 10: L123: The way this is written suggests that p value should be used as effect size. In any case, p values are not as important as effect sizes in this context. Please rewrite.

RESPONSE: We agree that p values should not be used as effect sizes. This sentence has been rewritten.

COMMENT 11: *L141-148, Fig. 3A-B. You would expect to see more biomass towards the bottom-right corner of the graphs if the effects of ND and FD just add up or if yield only increases when there “sufficient niche differentiation to offset fitness differences” (L84). Therefore, I do not see any reason to conclude in favor of the second alternative. A more sophisticated analysis would be required to tell between both hypotheses because their predictions are so similar (gradual increase towards the coexistence region of the graphs vs. a sharp increase at the divide between coexistence and exclusion). The pattern seems quite noisy to provide a clear cut between both patterns.

RESPONSE: We are grateful for these comments as they prompted us to conduct new analyses to test if it really is the excess of niche differences that matters or niche differences alone, see comment number 8. Moreover, we would like to mention that our hypothesis does not predict a sharp increase at the divide between coexistence and exclusion. Rather, it predicts a gradual increase in functioning from cases where small niche differences overcome small fitness differences to areas where niche differences become greater relative to the minimum necessary (the coexistence line). Our new analyses shows that the alternative hypothesis, that only niche differences contribute to high functioning, which would mean functioning increases with niche differences regardless of the size of the fitness differences, does not explain the pattern seen in Figure number 3.

COMMENT 12: L359: (20)?

RESPONSE: This reference has been included as suggested.

COMMENT 13: Equation after L359 (and elsewhere): When the experiment was set up, seeds of the focal and interacting species were added in different amounts measured as mass of seeds (2-16 g m⁻²), but the fitted model requires numbers (not mass) of individuals. Did you count the number of interacting individuals that got established in each plot, and used that number to fit your models? Please explain.

RESPONSE: We apologize for the lack of clarity. We counted the number of individuals of all species within a radius of 7.5 cm, which is a standard procedure of studies working with annual plant species. The number of individuals was used in our models. This is now explained in lines 417-418.

COMMENT 14: L360-61. I guess that the parentheses should close after “greater or equal to zero” rather than after “BFGS-B”).

RESPONSE: We have redone the estimation of competitive coefficients according to reviewer 1. We now just bound values of lambda to be greater than 1. This condition has therefore been changed.

COMMENT 15: L363: gi was calculated as the fraction of seeds that germinated out of the five that were sown in each plot, from the number of seeds added when setting the density treatment (those added as grams per m), or both?

RESPONSE: We have explained in careful detail how we measured species seed germination rates (g). See lines 411-414.

COMMENT 16: L389 and ss: I did not follow clearly how was litter manipulated and decomposition measured.

RESPONSE: We collected litter during the experiment while following litter senescence biweekly. Then, at the peak of leaf senescence, we put litter bags in the soil. We did not combine litter from several plots to ensure that the litter quality was obtained from the particular microenvironment of each plot. This is now explained in lines 460-468.

COMMENT 17: *L416 and ss: I like the idea of resorting to a procedure such as that proposed by Connolly et al. 2013 to solve the problem, and the fact that the procedure proposed here correlates with the “canonic” procedure of Loreau and Hector (although sometimes the correlation is quite weak, something that merits some comment from the authors) suggests that the Godoy et al.’s method used is appropriate. However, much more detail is required here. Connolly et al.’s procedure needs to be explained for the reader, and, very importantly, the way in which it was adapted/interpreted in terms of complementarity and selection needs to be thoroughly explained. In eq. 1 in Connolly’s paper, the model for no diversity effects is explained. I can but guess that the model used by Godoy et al is eq. 2b from Connolly, which is the same as eq 1 with only one more term comprising diversity effects ($\Delta \pi_j$ to the power theta). If I get it right, positive deltas result in overyielding and thus indicate complementarity, and theta is a shape parameter. I can’t see where the measurements of selection come from (unless theta can be interpreted in this way, but it is not obvious why). These things need to be explained, especially because Connolly et al. do not propose any interpretation of their model as a partition of diversity effects into complementarity and selection effects and thus Godoy et al.’s proposal seems to be entirely novel.

RESPONSE: We now explain in lines 488-512 how we interpret diversity interaction models following the Connolly et al. 2013 approach to compare it with the classic Loreau and Hector approach 2000. We basically used the equation 2a model in Connolly et al. 2013, and we interpret the intercepts as being measures of the selection effect because they quantify the overall effect of each species on each function. This should be analogous to the selection effect which measures the extent to which mixture functioning is driven by species with high or low monoculture functioning. The correlation is reasonable but note that the absolute values are higher, we rarely estimate negative selection with the Connolly et al. approach. However, this should not matter for linking to niche and fitness differences. We use the delta values (the pairwise interactions) as measures of the complementarity effects between species pairs. These analogies provided reasonable good correlations between both Connolly et al. 2013 and Loreau and Hector 2000 approaches, although we acknowledge that in some particular cases (soil N content under drought), the strength of the correlation is relatively low (see appendix S2).

COMMENT 18: L421: “because it possesses”

RESPONSE: Changed.

COMMENT 19: L428: Please provide a short explanation of what Loreau and Hector do, and why the results of your procedure needs to be compared with those derived from their method.

RESPONSE: This is now explained in lines 446-449.

COMMENT 20: L447: delete “this means that a”

RESPONSE: Deleted

COMMENT 21: L451: This is probably a personal bias, but I find the term “network analysis” a bit misleading here. What you are using here has long been known simply as stability analysis. Network analysis suggests to me the study of the structure of the matrix such as compartmentalization, connectivity, nestedness, small-world structure etc.

RESPONSE: We have removed the network analyses according to suggestion from reviewer 2.

COMMENT 22: Fig 1 “Non-linear instead regressions fitted the data better” remove “instead” and: better than what?

RESPONSE: Changed. Non-linear regressions fitted the data better than linear regressions.

COMMENT 23: *Fig 3: If there are no species pairs in the coexistence region under drought conditions (Figure S3), the complementarity effect observed in the coexistence region in panel B may be a strong extrapolation and thus may not be trusted. This would cast doubts on some important conclusions of the manuscript. Am I missing something?

RESPONSE: We have redone this figure to only show the extrapolation to where we observed niche and fitness differences, which for both treatments is around 0.7.

COMMENT 24: Fig S1: I am a bit surprised by the fact that the relationships between diversity and functioning are straight lines. Connolly et al model corresponds to a not linear relationship. What model are you using here?

RESPONSE: We are using the model 2A of the Connolly et al. 2013 approach. Fig. S1 shows the relationship between species richness and the net, selection and complementarity effects. Although the Connolly model assumes a non-linear relationship between biomass and species richness, we do not think there is a reason to expect a non-linear relationship between the diversity effects and species richness.

COMMENT 25: The caption of figure S5 states “see Figure S5”, which is odd. Is this an error and you want the reader to see some other figure instead? Besides, I find the figure unclear.

RESPONSE: Figure S5 has been deleted because network analyses are not presented in this revised version of the manuscript.

Reviewers' comments second round

Reviewer #2 (Remarks to the Author):

I have read the revised manuscript by Godoy and colleagues, and again I am struck by the elegance and importance of this paper. The authors have done an outstanding job responding to comments on the previous version.

I have one general thought, which may and may not be worth mentioning in the discussion part of the results & discussion section. Often there is an assumption that ecosystem function, as conceived from classic experiments, is tied to coexistence mechanisms or not impacted by diversity. But I think there is a third option, species contributions to function that do not necessarily depend on the underlying coexistence mechanism but could still be impacted by diversity. For example, pollination support where a species contribution to pollinator support would depend upon the relative dissimilarity to other community members, e.g., timing or colour or size, which is likely to be disconnected from coexistence. Take for example two communities completely structured by neutral dynamics, one has 4 grass species and the other has 3 grasses and 1 flowering forb. Pollination function would be much higher in the latter and the probability of including relatively unique species increases with diversity. All this to say, that there might be a subset of functions that are just never linked to coexistence.

Minor:

Fig. 1 and 2: increase the font size and symbol size. Also, avoid red and blue, because of colour blindness. Google "colour blind palette" for optimal colour matches.

Reviewer #3 (Remarks to the Author):

After reading the new version of ms NCOMMS-19-35084A I consider that most of my concerns have been satisfactorily addressed or rebutted. My overall evaluation of this contribution is still very positive. There is still one concern regarding my previous comment on whether the increase in function can be attributed to the additive, independent effects of stabilizing and equalizing mechanisms, or by their joint effect so that function increases with "excess niche differentiation". I like that the authors have resorted to the evidence and conducted new analyses, in this case Mantel tests. They find that niche differentiation is not as correlated with function as excess niche differentiation. Why not conduct an equivalent analysis using fitness differences vs. function? Assume you have two variables, one that is not correlated (niche differences) with a third (function), and a second (fitness differences) that is. Then, the sum (or difference, in this case excess differentiation) of both variables is expected to be correlated with the third. I would like to see if this is not the case.

Regarding the same issue, I like the new title "Ecosystem functioning is maximized when niche differences are larger than required for stable coexistence" better than the previous one. This is strictly true, although it still sounds to me like there is a qualitative divide between "larger than required" or "smaller", while we all concur that function increases gradually with stability (=excess differentiation, even if it is negative). I would prefer something that conveys the idea more clearly, although this is just a matter of opinion..

Response to Reviewers, third round -

RESPONSE TO REVIEWERS.

Reviewer #2

COMMENT 1: I have read the revised manuscript by Godoy and colleagues, and again I am struck by the elegance and importance of this paper. The authors have done an outstanding job responding to comments on the previous version.

I have one general thought, which may and may not be worth mentioning in the discussion part of the results & discussion section. Often there is an assumption that ecosystem function, as conceived from classic experiments, is tied to coexistence mechanisms or not impacted by diversity. But I think there is a third option, species contributions to function that do not necessarily depend on the underlying coexistence mechanism but could still be impacted by diversity. For example, pollination support where a species contribution to pollinator support would depend upon the relative dissimilarity to other community members, e.g., timing or colour or size, which is likely to be disconnected from coexistence. Take for example two communities completely structured by neutral dynamics, one has 4 grass species and the other has 3 grasses and 1 flowering forb. Pollination function would be much higher in the latter and the probability of including relatively unique species increases with diversity. All this to say, that there might be a subset of functions that are just never linked to coexistence.

RESPONSE: We thank the reviewer for these positive words. We also thank the reviewer for pointing out this critical comment. We agree with the reviewer that the drivers of species coexistence and functioning can be unrelated or weakly related. For the example the reviewer provides about pollination syndrome, we however think we could link species functioning and their maintenance by including the drivers explaining the simultaneous coexistence of the two trophic levels (plants and pollinators) This is what we were trying to say in lines 289-291. As this new perspective requires deep modeling and theoretical background we did not further explain it. Nevertheless, we now provide a sentence in line 292-294 including reviewer's comment.

Minor:

COMMENT 2: Fig. 1 and 2: increase the font size and symbol size. Also, avoid red and blue, because of colour blindness. Google "colour blind palette" for optimal colour matches.

RESPONSE: We have increased font size and changed color following the reviewer suggestion. New colors are blue and orange which are suggested by the color blind palette.

Reviewer #3

COMMENT 1: After reading the new version of ms NCOMMS-19-35084A I consider that most of my concerns have been satisfactorily addressed or rebutted. My overall evaluation of this contribution is still very positive. There is still one concern regarding my previous comment on whether the increase in function can be attributed to the additive, independent effects of stabilizing and equalizing mechanisms, or by their joint effect so that function increases with "excess niche differentiation". I like that the authors have resorted to the evidence and conducted new analyses, in this case Mantel tests. They find that niche differentiation is not as correlated with function as excess niche differentiation. Why not conduct an equivalent analysis using fitness differences vs. function? Assume you have two variables, one that is not correlated (niche differences) with a third (function), and a second (fitness differences) that is. Then, the sum (or difference, in this case excess differentiation) of both variables is expected to be correlated with the third. I would like to see if this is not the case.

RESPONSE: Following the reviewer comment, we have performed these additional analyses. Specifically, we have performed mantel tests correlating fitness differences alone with predicted pairwise functioning. These analyses revealed that pairwise fitness differences tend to negatively correlate with predicted biomass production (as expected) yet the correlations are not significant. Overall, these results suggest that both niche and fitness differences alone tend to correlate with functioning but its combination is a better predictor of functioning, at least of biomass production. These new analyses are included in lines 197-198. We have also modified the text in the introduction to prepare the reader that we evaluate the relationship with functioning of niche differences alone, fitness differences alone, or the combination by computing the excess of niche differences.

COMMENT 2: Regarding the same issue, I like the new title "Ecosystem functioning is maximized when niche differences are larger than required for stable coexistence" better than the previous one. This is strictly true, although it still sounds to me like there is a qualitative divide between "larger than required" or "smaller", while we all concur that function increases gradually with stability (=excess differentiation, even if

it is negative). I would prefer something than conveys the idea more clearly, although this is just a matter of opinion.

RESPONSE: We have changed the title to avoid this qualitative division. The new title we provide is the following: "An excess of niche differences maximizes ecosystem functioning". Therefore we avoid terms like "greater than" or larger than required".